# EPISODIC NOVELTY THROUGH TEMPORAL DISTANCE

**Yuhua Jiang**[1*]**, Qihan Liu**[1*]**, Yiqin Yang**[2†]**, Xiaoteng Ma**[1]**, Dianyu Zhong**[1]**, Hao Hu**[1]
**Jun Yang**[1†]**, Bin Liang**[1]**, Bo Xu**[2] **, Chongjie Zhang**[3]**, Qianchuan Zhao**[1†]
[1]Tsinghua University
[2]The Key Laboratory of Cognition and Decision Intelligence for Complex Systems,
  Institute of Automation, Chinese Academy of Sciences
[3]Washington University in St. Louis
{jiangyh22, lqh20}@mails.tsinghua.edu.cn

## ABSTRACT

Exploration in sparse reward environments remains a significant challenge in reinforcement learning, particularly in Contextual Markov Decision Processes (CMDPs), where environments differ across episodes. Existing episodic intrinsic motivation methods for CMDPs primarily rely on count-based approaches, which are ineffective in large state spaces, or on similarity-based methods that lack appropriate metrics for state comparison. To address these shortcomings, we propose **E**pisodic Novelty Through **T**emporal **D**istance (ETD), a novel approach that introduces temporal distance as a robust metric for state similarity and intrinsic reward computation. By employing contrastive learning, ETD accurately estimates temporal distances and derives intrinsic rewards based on the novelty of states within the current episode. Extensive experiments on various benchmark tasks demonstrate that ETD significantly outperforms state-of-the-art methods, highlighting its effectiveness in enhancing exploration in sparse reward CMDPs.

## 1 INTRODUCTION

Exploration in sparse reward environments remains a significant challenge in reinforcement learning (RL). Recent approaches have introduced the concept of intrinsic motivation (Meyer & Wilson, 1991; Oudeyer et al., 2007) to encourage agents to explore novel states, yielding promising results in sparse reward Markov Decision Processes (MDPs) (Bellemare et al., 2016; Pathak et al., 2017; Burda et al., 2018; Machado et al., 2020). Most existing methods grounded in intrinsic motivation derive rewards from the agent's cumulative experiences across all episodes. While these methods are effective in singleton MDPs, where agents are spawned in the same environment for each episode, they exhibit limited generalization across environments (Henaff et al., 2022). Real-world applications are often more suitably represented by Contextual MDPs (CMDPs) (Henaff et al., 2023), where different episodes correspond to different environments that nevertheless share certain characteristics, such as procedurally-generated environments (Chevalier-Boisvert et al., 2023; Cobbe et al., 2020; Küttler et al., 2020) or embodied AI tasks requiring generalization across diverse spaces (Savva et al., 2019; Li et al., 2021; Gan et al., 2020; Xiang et al., 2020). In CMDPs, the uniqueness of each episode indicates that experiences from one episode may offer limited insights into the novelty of states in another episode, thereby necessitating the development of more effective intrinsic motivation mechanisms.

To address the challenges of exploration in CMDPs, where episodes differ significantly, several works have introduced *episodic bonuses* (Henaff et al., 2023). These bonuses are derived from experiences within the current episode, avoiding the generalization limitations of cross-episode rewards. These approaches can typically be divided into two lines: count-based (Raileanu & Rocktäschel, 2020; Flet-Berliac et al., 2021; Zha et al., 2021; Zhang et al., 2021b; Parisi et al., 2021; Zhang et al., 2021a; Mu et al., 2022; Ramesh et al., 2022) and similarity-based (Savinov et al., 2018; Badia et al., 2020; Henaff et al., 2022; Wan et al., 2023). Count-based methods rely on an episodic count term to generate positive bonuses once encountering a new state but struggle in large or continuous state spaces (Lobel

---

*Equal contribution. Code is availabe at https://github.com/Jackory/ETD.
†Corresponding authors: yiqin.yang@ia.ac.cn, {yangjun603, zhaoqc}@tsinghua.edu.cn.

et al., 2023), where each state is unique and episodic bonuses remain uniform across all states. Meanwhile, similarity-based methods require appropriate measurements between pairs of states, which used to be assessed via Euclidean distance (Badia et al., 2020; Henaff et al., 2022) or reachable likelihood (Savinov et al., 2018; Wan et al., 2023) in some latent spaces. However, these similarity measurements used by existing methods do not provide a suitable metric for capturing the novelty of states, as illustrated in Figure 2. This inadequacy undermines the credibility of subsequent intrinsic reward calculations and limits the effectiveness of these methods in complex CMDP environments. Our work addresses this gap by introducing a new metric—temporal distance—that more effectively captures novelty in CMDPs by considering the expected number of steps between states.

In this work, we introduce **E**pisodic Novelty Through **T**emporal **D**istance (ETD), a novel approach designed to encourage agents to explore states that are temporally distant from their episodic history. The critical innovation of ETD lies in its use of *temporal distance*—the expected number of environment steps required to transition between two states—as a robust metric for state similarity in intrinsic reward computation. Unlike existing similarity metrics, temporal distance is invariant to state representations, which mitigates issues like the "noisy-TV" problem (Burda et al., 2018) and ensures the applicability of ETD in pixel-based environments. We employ contrastive learning with specialized parameterization to accurately estimate the temporal distances between states. The intrinsic reward is computed based on the aggregated temporal distances between a new state and each in the episodic memory. Through extensive experiments on various CMDP benchmark tasks, including MiniGrid (Chevalier-Boisvert et al., 2023), Crafter (Hafner, 2022), and MiniWorld (Chevalier-Boisvert et al., 2023), we show that ETD significantly outperforms state-of-the-art methods, improving exploration efficiency.

## 2 BACKGROUND

We consider a contextual Markov Decision Process (CMDP) defined by $(\mathcal{S}, \mathcal{A}, \mathcal{C}, P, r, \mu_C, \mu_S, \gamma)$, where $\mathcal{S}$ is the state space, $\mathcal{A}$ is the action space, $\mathcal{C}$ is the context space, $P : \mathcal{S} \times \mathcal{A} \times \mathcal{C} \to \Delta(\mathcal{S})$ is the transition function, $r(s_t, a_t, s_{t+1})$ is the reward function and typically sparse, $\mu_S$ is the initial state distribution conditioned on the context, $\mu_C$ is the context distribution, and $\gamma \in (0, 1)$ is the reward discount factor. At start at each episode, a context $c$ is sampled from $\mu_C$, followed by an initial state $s_0$ sampled from $\mu_S(\cdot|c)$, and subsequent states are sampled from $s_{t+1} \sim P(\cdot|s_t, a_t, c)$. The goal is to optimize a policy $\pi : \mathcal{S} \to \Delta(A)$ so that the the expected accumulated reward across over all contexts $\mathbb{E}_{c \sim \mu_C, s_0 \sim \mu_S(\cdot|c)}[\sum_t \gamma^t r(s_t, a_t, s_{t+1})]$ is maximized.

Examples of CMDPs include procedurally generated environments (Chevalier-Boisvert et al., 2023; Cobbe et al., 2020; Küttler et al., 2020; Hafner, 2022), where each context $c$ serves as a random seed for environment generation. Similarly, Embodied AI environments (Chevalier-Boisvert et al., 2023; Savva et al., 2019; Gan et al., 2020), where agents navigate various simulated homes, are also examples of CMDPs. Notably, singleton MDPs ($|C| = 1$) represent a special case of CMDPs. We primarily focus on CMDPs with $|C| = \infty$.

To address the sparse reward challenges, we augment the reward function $r$ by adding an intrinsic reward bonus. The modified equation is $r(s_t, a_t, s_{t+1}) = r_t^e + \beta \cdot b_t$, where $r_t^e$ represents the sparse extrinsic reward and $b_t$ denotes the intrinsic reward at each timestep $t$. The hyperparameter $\beta$ controls the influence of the intrinsic reward.

## 3 LIMITATIONS OF CURRENT EPISODIC BONUSES

Recent intrinsic motivation methods (Andres et al., 2022; Henaff et al., 2022; 2023) have demonstrated that incorporating an episodic bonus is crucial for solving sparse reward CMDP problems. For instance, high-performing methods like NovelD(Zhang et al., 2021b) often depend on an episodic count term to be effective in CMDPs. However, these count-based methods face challenges in large or noisy state spaces. When each state is unique, the episodic bonus becomes less meaningful as it assigns the same value to all states. This issue is evident in the "noisy-TV" problem (Burda et al., 2018), where random noise disrupts the state. We validated this observation using the MiniGird-DoorKey16x16 experiment, as shown in Figure 1. When no noise was present in the environment, NovelD performed well. However, when Gaussian noise with a mean of 0 and variance of 0.1 was added to the state input, NovelD failed completely, as indicated by the corresponding curve

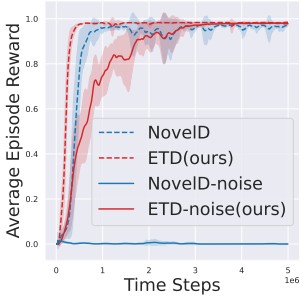

Figure 1: Training curves in Minigrid-DooKey-16x16 (w/w.o. noise).



(a) Inverse dynamic     (b) Likelihood     (c) Ours

Figure 2: Distance from ■ to all other states in a 17x17 SpiralMaze. Darker colors indicate greater distance. (**Left**) Euclidean distance of embeddings trained by inverse dynamics. (**Center**) Likelihood estimation of easy transitions (EC). (**Right**) The learned temporal distance (Ours).

(NovelD-noise). In contrast, our proposed method, ETD, maintained strong performance even with the added noise, demonstrating its robustness where NovelD proved ineffective.

Potential alternatives include computing episodic novelty based on the similarity of the current state to previously encountered states stored in episodic memory. This can be achieved using metrics such as Euclidean distance in an embedding space learned through inverse dynamics, as seen in NGU (Badia et al., 2020) and E3B (Henaff et al., 2022). Another approach is to estimate the likelihood of easy transitions between states, as done in EC (Savinov et al., 2018) and DEIR (Wan et al., 2023). However, we found that these methods do not accurately measure the similarity between states. We conducted an experiment using the SpiralMaze environment, as shown in the Figure 2. We collected 100 trajectories, each consisting of 50 time steps, as training data. We then evaluated the similarity between the red square state ■ and all other states. As the spiral deepens, the states become increasingly distant from the red square, indicating lower similarity. Figure 2(a) shows the inverse dynamics method, where embeddings are trained to predict actions based on the current and next states, with the Euclidean distance between embeddings serving as the similarity metric. However, we found that the Inverse Dynamics method struggles to provide a smooth estimation of the similarity between states. Figure 2(b) presents the EC approach, where a classifier predicts the likelihood that two states are within K time steps, using this likelihood as the similarity measure. We observed that this likelihood measure effectively captures local proximity but fails to distinguish the similarity between most states due to its nature as a non-distance metric. In contrast, our method, which learns temporal distances, as shown in Figure 2(c), accurately captures the similarity between the red square and each state.

## 4 METHODS

In this section, we introduce Episodic Novelty through Temporal Distance (ETD), an algorithm designed to enhance exploration in CMDPs. The core innovation of ETD is using a temporal distance quasimetric to measure state similarity, encouraging the agent to explore states that are temporally distant from its episodic memory. As summarized in Figure 3, we employ a specially parameterized contrastive learning method to learn this temporal distance and then identify the minimum temporal distance between the current state and all previously encountered states in the episodic memory. This minimum distance serves as the intrinsic reward for the current state. The next two sections will elaborate on the details.

### 4.1 TEMPORAL DISTANCE LEARNING

Temporal distance can be intuitively understood through the transition probability between states, where a lower probability indicates a larger distance. For a given policy $\pi$, we define $p^\pi(s_k = y|s_0 = x)$ as the probability of reaching state $y$ at time step $k$ when starting from $x$. The transition probability can be described using a discounted state occupancy measure, which equals a geometrically weighted average of the probabilities:

$$p_\gamma^\pi(s_f = y|s_0 = x) = (1-\gamma)\sum_{k=0}^{\infty}\gamma^k p^\pi(s_k = y|s_0 = x). \tag{1}$$

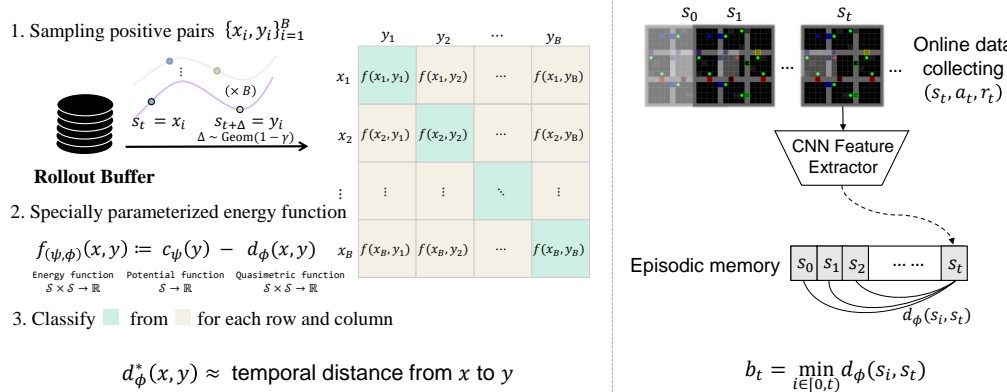

Figure 3: Overview of ETD. ETD encourages visits to temporally distant states from episodic memory. Temporal distance can be learned through contrastive learning when positive samples consist of a state and its geometrically distributed future state, with the erengy function parameterized as a potential network minus a quasimetric network. The intrinsic reward is then derived from the minimum temporal distance between the current state and the states stored in episodic memory.

To ensure the temporal distance behaves as a quasimetric (a metric that relaxes the symmetry assumption), we use the successor distance (Myers et al., 2024). Given a policy $\pi$, the successor distance is defined as the difference between the logarithms of the probabilities of reaching $y$ from $y$ (self-loop) and reaching $y$ from $x$:

$$d_{\text{SD}}^{\pi}(x,y) = \log\left(\frac{p_{\gamma}^{\pi}(s_f = y | s_0 = y)}{p_{\gamma}^{\pi}(s_f = y | s_0 = x)}\right). \tag{2}$$

This formulation satisfies the triangle inequality and other quasimetric properties (Myers et al., 2024), even in stochastic MDPs.

Consequently, the successor distance can be reliably used as a measure of similarity between states.

we present how to estimate the successor distance $d_{\text{SD}}^{\pi}(x,y)$ defined in Equation 2 based on the contrastive learning. Given the joint distribution $p_{\gamma}^{\pi}(s_f = y \mid s_0 = x)$ over two state $(x,y)$, define $p_s(x)$ as the marginal state distribution, and $p_{s_f}(y) = \int_s p_s(x) p_{\gamma}^{\pi}(s_f = y \mid s_0 = x)$ as the corresponding marginal distribution over future states. Intuitively, we can define an energy function $f_{\theta}(x,y)$ that assigns larger values to $(x,y)$ tuples sampled from the joint distribution and smaller values to $(x,y)$ tuples sampled independently from their marginal distributions. Give a batch of tuples $\{x_i, y_i\}_{i=1}^{B}$, we use the symmetrized infoNCE loss function (Oord et al., 2018) to learn $f_{\theta}(x,y)$:

$$\mathcal{L}_{\theta} = \sum_{i=1}^{B}\left[\log\frac{\exp(f_{\theta}(x_i, y_i))}{\sum_{j=1}^{B}\exp(f_{\theta}(x_i, y_j))} + \log\frac{\exp(f_{\theta}(x_i, y_i))}{\sum_{j=1}^{B}\exp(f_{\theta}(x_j, y_i))}\right]. \tag{3}$$

Based on Equation 3, we can use the unique solution of the energy function $f_{\theta^*}$ (Ma & Collins, 2018; Poole et al., 2019) to recover the successor distance $d_{\text{SD}}^{\pi}(x,y)$ in Equation 2. Following prior work (Myers et al., 2024), we decompose the energy function $f_{\theta=(\phi,\psi)}(x,y)$ as the difference between a potential network $c_{\psi}(y) : \mathcal{S} \to \mathbb{R}$ and a quasimetric network $d_{\phi}(x,y) : \mathcal{S} \times \mathcal{S} \to \mathbb{R}$:

$$f_{\theta=(\phi,\psi)}(x,y) = c_{\psi}(y) - d_{\phi}(x,y). \tag{4}$$

Then, we have $d_{\text{SD}}^{\pi}(x,y) = d_{\phi^*}(x,y)$. As a result, we discard $c_{\psi(y)}$ after contrastive learning and directly use $d_{\phi}(x,y)$ as our temporal distance. For further details, see Appendix B.

To demonstrate the learned temporal distance, we present results from the SpiralMaze 17x17 task, as shown in Figure 2(c). We collected 100 random episodes (each with 50-time steps) and minimized the loss function following the process above. The resulting temporal distance $d_{\phi}(\blacksquare, \cdot)$ is visualized with a colormap (darker color indicates larger distances), showing strong alignment with ground-truth.

---

**Algorithm 1** Episodic Novelty through Temporal Distance

Initialize policy $\pi$, quasimetric $d_\phi$, potential $c_\psi$ and $f_{(\phi,\psi)} = c_\psi - d_\phi$.
**while** not converged **do**
    Sample context $c \sim \mu_C$ and initial state $s_0 \sim \mu_S(\cdot|c)$
    **for** $t = 0, ..., T$ **do**
        $a_t \sim \pi(\cdot|s_t)$         # Sample action
        $s_{t+1}, r^e_{t+1} \sim P(\cdot|s_t, a_t, c)$    # Step through environment
        $b_{t+1} = \min_{k \in [0,t+1]} d_\phi(s_k, s_{t+1})$  # Compute bonus
        $r_{t+1} = r^e_{t+1} + \beta b_{t+1}$
    **end for**
    Sample pair of states $\{(x_i, y_i)\}^B_{i=1} \sim p^\pi_\gamma(s^f = y_i|s_0 = x_i)p_s(x_i)$
    # Practically, $x_i = s_t, y_i = s_{t+j}, j \sim \text{Geom}(1-\gamma)$.
    Update $f_{(\phi,\psi)}$ to minimize the loss:

$$\mathcal{L}_{(\phi,\psi)} = \sum_{i=1}^{B} \left[ \log \frac{\exp(f_{(\phi,\psi)}(x_i, y_i))}{\sum_{j=1}^{B} \exp(f_{(\phi,\psi)}(x_i, y_j))} + \log \frac{\exp(f_{(\phi,\psi)}(x_i, y_i))}{\sum_{j=1}^{B} \exp(f_{(\phi,\psi)}(x_j, y_i))} \right]$$

    Perform PPO update on $\pi$ using rewards $r_1, ..., r_T$.
**end while**

---

## 4.2 TEMPORAL DISTANCE AS EPISODIC BONUS

Our approach maximizes the temporal distance between newly visited and previously encountered states within the current episode. At each time step $t$, we assign a larger intrinsic reward to states that are temporally distant from the episodic memory. Formally, the episodic temporal distance bonus is defined as:

$$b_{\text{ETD}}(s_t) = \min_{k \in [0,t)} d_\phi(s_k, s_t), \tag{5}$$

where $\{d_\phi(s_k, s_t)\}^{t-1}_{k=0}$ represents the learned temporal distances between the current state $s_t$ and all previous states $\{s_k\}^{t-1}_{k=0}$ in the episodic memory. The minimum distance is used as the episodic intrinsic reward. In terms of computational efficiency, storing CNN-extracted embeddings in episodic memory minimizes memory overhead. Additionally, concatenating memory states allow all temporal distances to be computed in a single neural network inference, ensuring high time efficiency.

**Connections to previous intrinsic motivation methods.** Many previous episodic intrinsic reward methods, such as DEIR (Wan et al., 2023), NGU (Badia et al., 2020), GoBI (Fu et al., 2023), and EC (Savinov et al., 2018), also rely on episodic memory and past states to calculate rewards. Compared to these methods, our reward formulation is notably simpler. Both EC and GoBI use reachability to assess state similarity, which is similar to our approach. However, EC struggles to learn temporal distance accurately, as shown in Figure 2(b). Meanwhile, GoBI depends on a world model's lookahead rollout to estimate temporal distance, which results in high computational complexity.

## 4.3 FULL ETD ALGORITHMS

We implement our quasimetric network $d_\phi$ using MRN (Liu et al., 2023), which allows us to generate an asymmetric distance. Detailed structural information about the network is provided in the Appendix E.1.2. The potential network $c_\psi$ is a multi-layer perceptron (MLP) that shares the same CNN backbone as the MRN. The complete algorithm is outlined in Algorithm 1.

## 5 EXPERIMENTS

To evaluate the capabilities of existing methods and assess ETD, we aim to identify CMDP environments that present challenges typical of realistic scenarios, such as sparse rewards, noisy or irrelevant features, and large state spaces. We consider three domains, including the Minigrid (Chevalier-Boisvert et al., 2023) and its noisy variants, as well as high-dimensional pixel-based Crafter (Hafner,

2022) and Miniworld (Chevalier-Boisvert et al., 2023). For all the experiments, we use PPO as the base RL algorithm and add intrinsic rewards specified by various methods to encourage exploration.

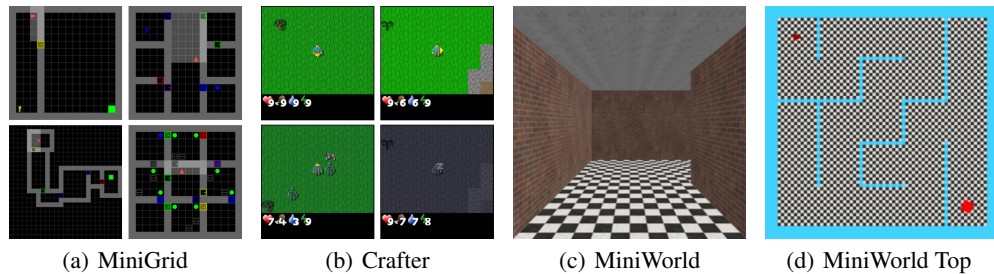

|        (a) MiniGrid        |       (b) Crafter        |       (c) MiniWorld       |     (d) MiniWorld Top     |

Figure 4: Rendering of the environments used in this work.

We compare ETD against 7 previous methods, including DEIR (Wan et al., 2023), NovelD (Zhang et al., 2021b), E3B (Henaff et al., 2022), EC (Savinov et al., 2018), NGU (Badia et al., 2020), RND (Burda et al., 2018). We also included a pure episodic count-based method named Count, which recent studies (Henaff et al., 2023) have shown to be a strong baseline in CMDPs. We carefully tuned the hyperparameters of each method through grid search to ensure optimal performance.

## 5.1 MINIGRID ENVIRONMENTS

MiniGrid (Chevalier-Boisvert et al., 2023) features procedurally generated 2D environments tailored for challenging exploration tasks. In these environments, agents interact with objects such as keys, balls, doors, and boxes while navigating multiple rooms to locate a randomly placed goal. The agents receive a single sparse reward upon completing each episode. We chose four particularly challenging environments: MultiRoom, DoorKey, KeyCorridor, and ObstructedMaze. In the MultiRoom environment, the agent's task is relatively straightforward, requiring navigating through a series of interconnected rooms to reach the goal. DoorKey presents an increased difficulty, as the agent must first find and pick up a key and then open a door before reaching the goal. KeyCorridor is even more demanding, requiring the agent to open multiple doors, locate a key, and then use it to unlock another door to access the goal. ObstructedMaze is the most complex of all: the key is hidden within a box, a ball obstructs the door, and the agent must find the hidden key, move the ball, open the door, and finally reach the goal. Further details on these tasks can be found in the Appendix.

Figure 5 illustrates the learning curves of ETD and other state-of-the-art exploration baselines across eight challenging MiniGrid navigation tasks. Notably, MultiRoom-N6, DoorKey-16x16, and KeyCorridor-S6R3 are the most challenging settings for their respective environments, while ObstructedMaze offers multiple difficulty levels, from 2Dlh to the most challenging Full version. ETD consistently outperforms all other methods, especially in the ObstructedMaze-Full environment, where it reaches near-optimal performance within 20 million steps, demonstrating double the sample efficiency compared to NovelD, the strongest baseline. Our implementation of NovelD achieved the highest reported performance in the literature, highlighting the careful tuning of our baseline.

From Figure 5, we can observe that PPO fails to learn effectively without intrinsic reward. Additionally, the global intrinsic reward method RND is almost ineffective, which might be due to the ineffectiveness of global exploration experience in CMDP. We also notice that NGU with episodic intrinsic reward performs poorly, possibly because NGU is primarily designed for Atari tasks, with its episodic intrinsic reward tailored for that domain. NovelD, on the other hand, performs quite well, largely due to its episodic count mechanism, and the global bonus in the current NovelD also contributes to its performance. Meanwhile, Count, which only uses the episodic count of NovelD as the reward, achieves relatively good results on certain maps, such as Obstructed-2Dlh, but it still significantly lags behind NovelD. Pure episodic intrinsic reward methods like DEIR, E3B, and EC all perform relatively well, but their similarity measurements lack precision. In contrast, our ETD method, which leverages temporal distance, significantly improves exploration learning efficiency. We did not compare it with GoBI because it requires pretraining a world model, and the rollout costs of the world model are pretty high. Moreover, we observed that the convergence rate of our learning curves is already significantly faster than what was demonstrated in the GoBI paper.

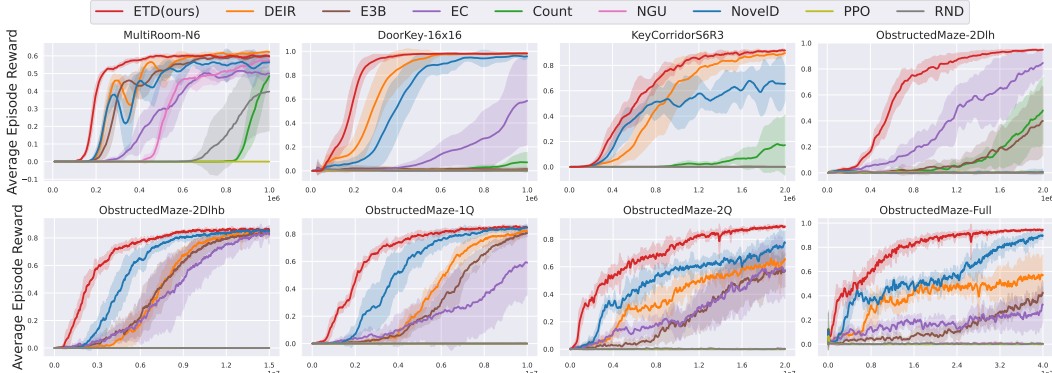

Figure 5: Training performance of ETD and the baselines on 8 most challenging Minigrid environments. The x-axis represents the environment steps. All the results are averaged across 5 seeds.

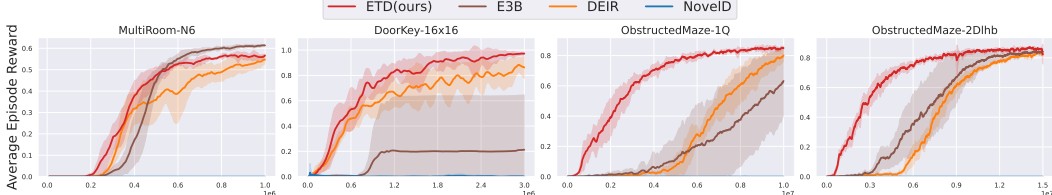

Figure 6: Training performance on Minigrid with noise environments. The x-axis represents the environment steps. All results are averaged across 5 seeds.

## 5.2 MINIGRID ENVIRONMENTS WITH NOISE

To better simulate realistic scenarios, we introduced noise into the states of MiniGrid, resulting in stochastic dynamics and ensuring that no two states are identical. The noise is generated as Gaussian noise with a mean of 0 and a variance of 0.1, which is then directly added to the states. We compared the ETD method with three effective methods for MiniGrid: DEIR, NovelD, and E3B. The results are presented in Figure 6.

Our results indicate that NovelD, a count-based method, completely failed to effectively guide exploration, as the episodic rewards based on counts no longer provided useful information. In contrast, similarity-based methods such as E3B and DEIR continued to perform reasonably well. However, our approach provided a more accurate assessment of state similarity by utilizing temporal distance. Even in the presence of noise, temporal distance effectively represented the similarity between two states, while the inverse dynamics representation learning used in E3B and the discriminative representation learning used in DEIR could not perfectly measure the distance between states, allowing our method to outperform both E3B and DEIR.

## 5.3 ABLATIONS OF ETD

**Representation Learning** To further illustrate the effectiveness of temporal distance as an intrinsic reward, we compare the ETD with the Euclidean distance within both inverse dynamics and discriminator representation learning contexts. Discriminator representation learning, introduced in DEIR, resembles contrastive learning and predicts whether two states and an action are part of a truly observed transition. While all these techniques utilize ETD as a form of intrinsic reward, they differ in evaluating similarities between states. The results of comparisons are illustrated in Figure 7. In the Doorkey-16x16 task, the performance difference is not significant. However, in the ObstructedMaze-1Q task, where the state

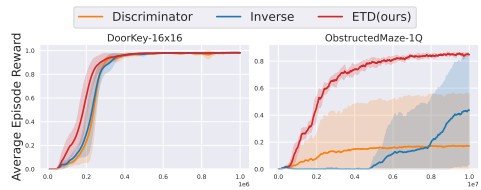

Figure 7: Ablation of representation learning.

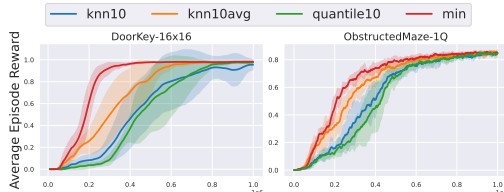
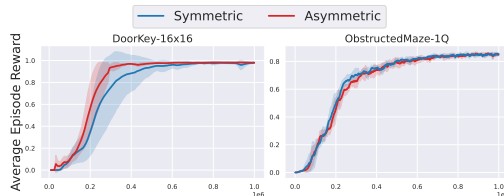

Figure 8: Ablation of aggregate function      Figure 9: Ablation of Asymmetric / Symmetric

is considerably richer, ETD outperforms both the inverse dynamic and discriminator methods. This finding indicates that a more accurate distance measurement contributes significantly to exploration efficiency.

**Aggregate function** For the intrinsic reward formulation, we consider not only the minimum but also other functions, such as the 10% quantile (quantile10), the 10th nearest neighbor (knn10), and the average of the 1st to 10th nearest neighbors (knn10avg). The comparisons are presented in Figure 8. We observe that the minimum consistently outperforms the other functions. This is because the minimum provides the most aggressive reward signal. For example, if a state in the episodic memory matches the current state, the minimum yields a reward signal of zero. This aggressive reward discourages the agent from revisiting similar states, thus enhancing exploration efficiency.

**Symmetric** Our ETD method uses a quasimetric distance function, which is inherently asymmetric. However, symmetric alternatives can also be considered. For instance, by removing the asymmetric components from the MRN, we can obtain a symmetric distance function. The comparison results are shown in Figure 9. Interestingly, the performances of both the asymmetric and symmetric versions are nearly identical. Given that most environment transitions exhibit more symmetry than asymmetry, employing a symmetric distance function is reasonable. Nevertheless, to retain the generality of our approach, we choose an asymmetric distance function as the default.

### 5.4 Pixel-Based Crafter and MiniWorld Maze

To evaluate the scalability of our method to continuous high-dimensional pixel-based observations, we conducted experiments on two pixel-based CMDP benchmarks: Crafter and MiniWorld.

Crafter (Hafner, 2022) is a 2D environment with randomly generated worlds and pixel-based observations (64x64x3), where players complete tasks such as foraging for food and water, building shelters and tools, and defending against monsters to unlock 22 achievements. The reward system is sparse, granting +1 for each unique achievement unlocked per episode and a -0.1/+0.1 reward based on life points. With a budget of 1 million environmental steps, Crafter suggests evaluating performance using both the success rate of 22 achievements and a geometric mean score, which we adopt as our performance metric. Additionally, we conducted experiments without life rewards, as they often hindered learning efficiency.

MiniWorld (Chevalier-Boisvert et al., 2023) is a procedurally generated 3D environment simulator that offers a first-person, partially observable view as the primary form of observation. We focused on the MiniWorld-Maze, where the agent must navigate through a procedurally generated maze. Exploration in this environment is particularly challenging due to the 3D first-person perspective and the limited field of view. Additionally, no reward is given if the agent fails to reach the goal within the time limit, further increasing the difficulty.

We compared ETD against DEIR, NovelD, and PPO without intrinsic rewards. As illustrated in Figure 10 and Figure 11, ETD consistently outperformed or matched the baseline algorithms, demonstrating its superior ability to address CMDP challenges with high-dimensional pixel-based observations.

## 6 Related Work

**Intrinsic Motivation in RL** Exploration driven by intrinsic motivation has long been a key focus in the RL community (Oudeyer et al., 2007; Oudeyer & Kaplan, 2007). Various methods that

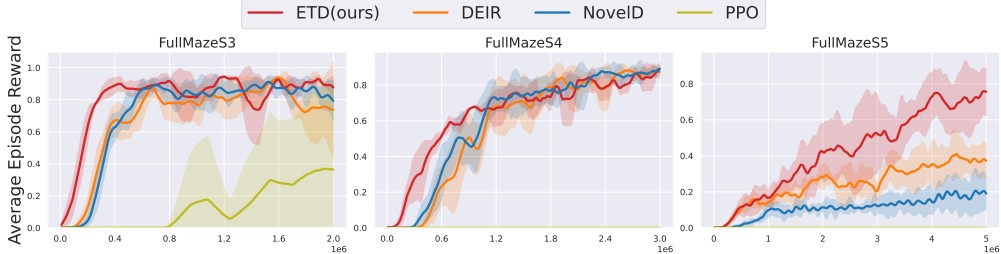

Figure 10: Training performance of ETD and the baselines on MiniWorld Maze with different sizes.

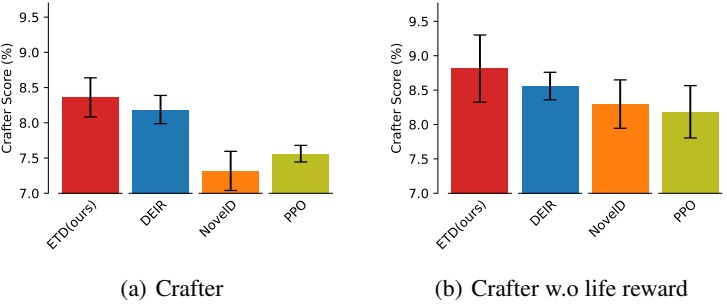

(a) Crafter       (b) Crafter w.o life reward

Figure 11: Evaluating ETD and the baselines on Crafter.

combine deep RL agents with exploration bonuses have been developed. Notable examples include ICM (Pathak et al., 2017), RND (Burda et al., 2018), and pseudocounts (Bellemare et al., 2016; Martin et al., 2017; Ostrovski et al., 2017; Machado et al., 2020), which have demonstrated success in challenging tasks like Montezuma's Revenge (Bellemare et al., 2013). These methods, often categorized as global bonus approaches, were primarily designed for singleton MDPs, presenting limitations in CMDPs, where environments vary across episodes (Savva et al., 2019; Li et al., 2021; Gan et al., 2020; Xiang et al., 2020; Zisselman et al., 2023; Jiang et al., 2024).

To address this limitation, recent works have proposed episodic bonuses (Henaff et al., 2023) relying on episodic memory (Blundell et al., 2016; Pritzel et al., 2017), where intrinsic rewards are derived from experiences within the current episode. These methods can be roughly grouped into two categories: count-based (Raileanu & Rocktäschel, 2020; Flet-Berliac et al., 2021; Zha et al., 2021; Zhang et al., 2021b; Parisi et al., 2021; Zhang et al., 2021a; Mu et al., 2022; Ramesh et al., 2022) and similarity-based (Savinov et al., 2018; Badia et al., 2020; Henaff et al., 2022; Wan et al., 2023). Combining global and episodic bonuses and effectively utilizing both remains an open challenge. Approaches like AGAC (Flet-Berliac et al., 2021), RIDE (Raileanu & Rocktäschel, 2020), and NovelD (Zhang et al., 2021b) utilize both, yielding better performance in CMDPs (Parisi et al., 2021; Zhang et al., 2021a; Mu et al., 2022; Ramesh et al., 2022). However, other methods, such as EC (Savinov et al., 2018) and E3B (Henaff et al., 2022), focus solely on episodic bonuses and have also succeeded in CMDPs. Our approach belongs to the latter category, leveraging episodic bonuses to enhance performance in CMDPs. Table 1 compares recent intrinsic motivation approaches and highlights our method.

Our approach, which employs temporal distance as an intrinsic reward, shares similar ideas with EC (Savinov et al., 2018) and GoBI (Fu et al., 2023). EC also utilizes contrastive learning to assess the temporal proximity of states. However, while EC only predicts the probability that two states are temporally close, our method defines temporal distance as a theoretically quasimetric measure. GoBI uses a learned world model and extensive random rollouts to simulate reachable states, rewarding uniqueness. However, GoBI requires world model pretraining and incurs substantial computational costs. In contrast, our method achieves comparable performance while maintaining lower computational overhead.

**Temporal Distance in RL**    Temporal distance has been extensively applied in imitation learning (Sermanet et al., 2018), unsupervised reinforcement learning (Park et al., 2024; Hartikainen et al.,

| Method | Intrinsic Bonus: $b_{\text{Method}}(s_t)$ | Episodic Bonus Category |
|---|---|---|
| ICM | $\|\hat{\phi}(s_t) - \phi(s_t)\|_2^2$ | / |
| RND | $\|f(s_t) - \bar{f}(s_t)\|_2^2$ | / |
| AGAC | $D_{\text{KL}}\left(\pi\left(\cdot \mid s_t\right) \| \pi_{\text{adv}}\left(\cdot \mid s_t\right)\right) + \beta \cdot \frac{1}{\sqrt{N_e(s_{t+1})}}$ | Count |
| RIDE | $\|\phi\left(s_{t+1}\right) - \phi\left(s_t\right)\|_2 \cdot \frac{1}{\sqrt{N_e(s_t)}}$ | Count |
| NovelD | $[b_{\text{RND}}(s_{t+1}) - b_{\text{RND}}(s_t)]_+ \cdot \mathbb{I}[N_e(s_t) = 1]$ | Count |
| NGU | $b_{\text{RND}}(s_t) \cdot \frac{1}{\left(\sqrt{\sum_{\phi_i \in N_k} K(\phi(s_t), \phi_i) + c}\right)}$ | Similarity |
| E3B | $\phi\left(s_t\right)^\top \left[\sum_{i=0}^{t-1} \phi\left(s_i\right) \phi\left(s_i\right)^\top + \lambda I\right]^{-1} \phi\left(s_t\right)$ | Similarity |
| EC | $\alpha(\beta - F\{C(s_i, s_t)\}_{i \in |M|})$ | Similarity |
| DEIR | $\min_{i \in |M|}\left\{\frac{\|\phi(s_i), \phi(s_t)\|^2}{\|\phi_{\text{rnn}}(s_i), \phi_{\text{rnn}}(s_t)\|}\right\}$ | Similarity |
| **ETD(ours)** | $\min_{i \in |M|} d_{\text{SD}}(s_i, s_t)$ | Similarity |

Table 1: Summary of recent intrinsic motivation methods. We marked the episodic bonus as Blue.

2019; Klissarov & Machado, 2023), and goal-conditioned reinforcement learning (Kaelbling, 1993; Durugkar et al., 2021; Eysenbach et al., 2022; Wang et al., 2023). Common methods for learning temporal distance include Laplacian-based representations (Wu et al., 2019; Wang et al., 2021; 2022), which use spectral decomposition to capture the geometry of the state space; constrained optimization (Park et al., 2024; Wang et al., 2023), which maintains a distance threshold between adjacent states while dispersing others; and temporal contrastive learning (Sermanet et al., 2018; Eysenbach et al., 2022), which brings temporally close states together in representation space while pushing apart negative samples. Each approach, however, has its limitations: Laplacian-based representations can be unstable during training (Gomez et al., 2024), constrained optimization highly depends on deterministic environments (Wang et al., 2023), and temporal contrastive learning often violates the triangle inequality (Wang et al., 2023), a key property of metrics.

Recently, CMD (Myers et al., 2024) proposes the successor distance, which theoretically guarantees a quasimetric temporal distance by using a specific parameterization of temporal contrastive learning. While CMD is limited to goal-conditioned tasks, we extend this method to sparse reward CMDPs.

# 7 CONCLUSION

In this work, we introduce ETD, a novel episodic intrinsic motivation method for CMDPs. ETD leverages temporal distance as a measure of state similarity, which is more robust and accurate than previous methods. This allows for more effective calculation of intrinsic rewards, guiding agents to explore environments with sparse rewards. We demonstrate that ETD significantly outperforms existing episodic intrinsic motivation methods in sample efficiency across various challenging domains, establishing it as the state-of-the-art RL approach for sparse reward CMDPs.

# ACKNOWLEDGMENTS

This work was supported by National Natural Science Foundation of China under Grant No. 62192751, in part by the National Science and Technology Innovation 2030 - Major Project (Grant No. 2022ZD0208804), in part by Key R&D Project of China under Grant No. 2017YFC0704100, the 111 International Collaboration Program of China (No.B25027) and in part by the BNRist Program under Grant No. BNR2019TD01009, in part by the InnoHK Initiative, The Government of HKSAR; and in part by the Laboratory for AI-Powered Financial Technologies.

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

## A LIMITATIONS

Using reward bonuses, whether within an episode or globally, violates the MDP assumption where the reward $r_t$ depends only on the immediately preceding state $s_t$ and action $a_t$. Even if the original RL task is an MDP, introducing a reward bonus that depends on the episode history implicitly transforms the task into a POMDP setting. This transformation is problematic because the value function based on the Markovian state might be biased. Although reward bonuses have proven highly effective for exploration in sparse reward settings, further research is needed to mitigate the impact of the POMDP transformation. Additionally, the successor distance we used may be infinite in non-ergodic settings, implying an assumption that we're dealing with ergodic MDP problems.

Another limitation of this work is that, our intrinsic reward is purely episodic and doesn't incorporate global bonuses from all experiences, limiting its application in singleton MDPs where global bonuses are essential. We believe designing a combined approach using temporal distance for both global and episodic bonuses is a promising direction. A potential method could involve designing a global bonus through maximum state entropy Seo et al. (2021); Liu & Abbeel (2021); Bae et al. (2024) and integrating it with ETD. We leave this exploration for future work.

## B THEORETICAL PROPERTIES OF SUCCESSOR DISTANCE

Here we list the most relevant properties of successor distance (Myers et al., 2024), which we used in the this paper as the temporal distance. The definition of successor distance (Myers et al., 2024) is independent of $\pi$, whereas our successor distance is dependent of $\pi$ and learned in an on-policy manner. We also provide proof that our on-policy form of successor distance still satisfies the quasimetric property, which differs from (Myers et al., 2024).

**Proposition 1.** *For all $\pi \in \Pi$, $x, y \in S$, define the random variable $H^\pi(x, y)$ as the smallest transit time from $x$ to $y$, i.e., the hitting time of $y$ from $x$,*

$$d_{SD}^\pi(x, y) = -\log \mathbb{E}\left[\gamma^{H^\pi(x,y)}\right].$$

*Proof.* Starting from state $x$ and given $H^\pi(x, y) = h$), let $p^\pi(s_t = y | s_0 = x, H^\pi(x, y) = h)$ denotes the probability of reaching state $y$ at the time $t$, we have

$$p^\pi(s_t = y | s_0 = x, H^\pi(x, y) = h) = \begin{cases} 0 & \text{if } t < h \\ p^\pi(s_t = y | s_h = y) & \text{if } t \geq h \end{cases}. \tag{6}$$

And thus,

$$
\begin{aligned}
p_\gamma^\pi(s_f = y \mid s_0 = x) &= (1 - \gamma) \sum_{t=0}^\infty \gamma^t p^\pi(s_t = y \mid s_0 = x) \\
&= (1 - \gamma) \sum_{t=0}^\infty \sum_{h=0}^\infty \gamma^t p^\pi(s_t = y \mid s_0 = x, H^\pi(x, y) = h) P(H^\pi(x, y) = h) \\
&= (1 - \gamma) \sum_{h=0}^\infty p^\pi(H^\pi(x, y) = h) \sum_{t=0}^\infty \gamma^t P(s_t = y \mid s_0 = x, H^\pi(x, y) = h) \\
&= (1 - \gamma) \sum_{h=0}^\infty p^\pi(H^\pi(x, y) = h) \sum_{t=h}^\infty \gamma^t p_\gamma^\pi(s_t = y \mid s_0 = y) \\
&= \sum_{h=0}^\infty \gamma^h P(H^\pi(x, y) = h) \left((1 - \gamma) \sum_{t=h}^\infty \gamma^{t-h} p^\pi(s_t = y \mid s_0 = y)\right) \\
&= \sum_{h=0}^\infty \gamma^h P(H^\pi(x, y) = h) \left((1 - \gamma) \sum_{t=0}^\infty \gamma^t p^\pi(s_t = y \mid s_0 = y)\right) \\
&= \mathbb{E}_{H^\pi(x,y)}\left[\gamma^{H^\pi(x,y)}\right] p_\gamma^\pi(s_t = y \mid s_0 = y).
\end{aligned}
\tag{7}
$$

Therefore,

$$d_{\text{SD}}^{\pi}(x, y) = \log \left( \frac{p_{\gamma}^{\pi}(s_f = y | s_0 = y)}{p_{\gamma}^{\pi}(s_f = y | s_0 = x)} \right) = -\log \mathbb{E} \left[ \gamma^{H^{\pi}(x,y)} \right]. \tag{8}$$

$\square$

**Corollary 1.** *Assume $H^{\pi}(x, y)$ is a deterministic value,*

$$d_{SD}^{\pi}(x, y) = c \cdot H^{\pi}(x, y), \text{ where c is a free value.}$$

*Proof.* Following Proposition 1,

$$d_{\text{SD}}^{\pi}(x, y) = -\log \gamma^{H^{\pi}(x,y)} = H^{\pi}(x, y) \cdot \log \frac{1}{\gamma}. \tag{9}$$

$\square$

**Proposition 2.** *$d_{SD}$ is a quasimetric over S, satisfying the Positivity, Identity and triangle inequality.*

*Proof.* A distance function $d : \mathcal{S} \times \mathcal{S} \to \mathcal{R}$ is called quasimetric if it satisfies the following for any $x, y, z \in \mathcal{S}$.

1. Positivity: $d(x, y) \geq 0$

2. Identity: $d(x, y) = 0 \Leftrightarrow x = y$

3. Triangle inequality: $d(x, z) \leq d(x, y) + d(y, z)$

**Positivity:** From Proposition 1, we have $d_{\text{SD}} = -\log \mathbb{E}_{H^{\pi}(x,y)} \left[ \gamma^{H^{\pi}(x,y)} \right] \geq 0$.

**Identity:**

- $\Rightarrow$: $d_{\text{SD}}^{\pi}(x, y) = 0$ if and only if $p_{\gamma}^{\pi}(s_f = y | s_0 = x) = p_{\gamma}^{\pi}(s_f = y | s_0 = y)$, which occurs when $x = y$. For $x \neq y$, $H^{\pi}(x, y) \geq 1$, so by Proposition 1, $d_{\text{SD}}^{\pi}(x, y) \geq \log \frac{1}{\gamma}$.

- $\Leftarrow$: When $x = y$, $p_{\gamma}^{\pi}(s_f = y | s_0 = x) = p_{\gamma}^{\pi}(s_f = y | s_0 = y)$, thus $d_{\text{SD}}^{\pi}(x, y) = 0$.

**Triangle Inequality:** According to (Hunter, 2005) (Lemma 4.1), the hitting time $H^{\pi}(x, y)$ satisfies the triangle inequality, that is, $H^{\pi}(x, y) \leq H^{\pi}(x, z) + H^{\pi}(y, z)$. Let $f(H^{\pi}(x, y)) = -\log \mathbb{E}[\gamma^{H^{\pi}(x,y)}]$, $\log \mathbb{E}[\gamma^{H^{\pi}(x,y)}]$ is a convex function, and thus $f$ is a concave function. Furthermore, $f(0) = 0$. By the property of concave functions (ictibones, 2017), $f$ is subadditive, i.e., $f(a + b) \leq f(a) + f(b)$ for all $a$ and $b$. As desired, $f(H^{\pi}(x, y)) \leq f(H^{\pi}(x, z) + H^{\pi}(y, z)) \leq f(H^{\pi}(x, z)) + f(H^{\pi}(y, z))$, and thus , $d_{\text{SD}}^{\pi}(x, y) = f(H^{\pi}(x, z))$ satisfying the triangle inequality.

$\square$

**Proposition 3.** *For $x \neq y$, the unique solution to the the loss function in Equation 3 with the parametrization in Equation 4 is*

$$d_{\phi^*}(x, y) = \log \left( \frac{p_{\gamma}^{\pi}(s_f = y | s_0 = y)}{p_{\gamma}^{\pi}(s_f = y | s_0 = x)} \right).$$

*Proof.* If the batch size is large enough, the optimal energy function in Equation 3 satisfy

$$f_{\theta}^*(x, y) = \log \left( \frac{p_{\gamma}^{\pi}(s^f = y | s_0 = x)}{C \cdot p_{s_f}(y)} \right), \text{ where C is a free value.} \tag{10}$$

We can further decompose the optimal function into a potential function that depends solely on the future state minus the successor distance function,

$$f_\theta^*(x, y) = \underbrace{-\log\left(\frac{p_\gamma^\pi(s^f = y | s_0 = y)}{C \cdot p_{s_f}(y)}\right)}_{c_\psi(y)} \underbrace{-\log\left(\frac{p_\gamma^\pi(s_f = y | s_0 = y)}{p_\gamma^\pi(s_f = y | s_0 = x)}\right)}_{d_\phi(x,y)}. \tag{11}$$

$$\begin{aligned}
\log\left(\frac{p_\gamma^\pi(s_f = y | s_0 = y)}{p_\gamma^\pi(s_f = y | s_0 = x)}\right) &= f_\theta^*(y, y) - f_\theta^*(y, x) \\
&= c_\psi^*(y) - d_\phi^*(y, y) - (c_\psi^*(y) - d_\phi^*(x, y)) \\
&= d_\phi^*(x, y))
\end{aligned}$$

$\square$

# C  ADDITIONAL EXPERIMENTS

## C.1  EXPERIMENTS IN CONTINUOUS ACTION SPACE

We further conduct experiments in DeepMind Control (Tassa et al., 2018) (DMC) and MetaWorld (Yu et al., 2020) and HalfCheetahVelSparse.

**DMC**   We selected three tasks with relatively sparse rewards from the DMC environment: Acrobot-swingup_sparse, Hopper-hop, and Cartpole-swingup_sparse. As shown in Fig. 12, our results demonstrate that ETD consistently performs well, showing significant improvements over PPO. We also reproduced two baselines that excelled in discrete tasks—NovelD and E3B—but found they couldn't maintain consistent performance across all three continuous control tasks.

**MetaWorld**   Meta-World environments typically use dense rewards. We modified these to provide rewards only when tasks succeed. We tested on Reach, Hammer, and Button Press tasks, finding that PPO without intrinsic rewards performed well. This suggests that Meta-World may not be ideal benchmarks for exploration problems. To our knowledge, intrinsic motivation methods haven't been extensively tested on Meta-World, making it less suitable for comparing exploration-based methods.

**HalfCheetahVelSparse**   We modified the Mujoco HalfCheetah environment's forward reward function to provide rewards only when the target velocity is reached. Our experiments revealed that PPO already performs well in this task. Adding intrinsic motivation methods didn't significantly improve performance, likely due to the low exploration difficulty. These environments require relatively low exploration capabilities, do not necessitate intrinsic motivation methods, and are unsuitable as benchmarks for comparing exploration methods.

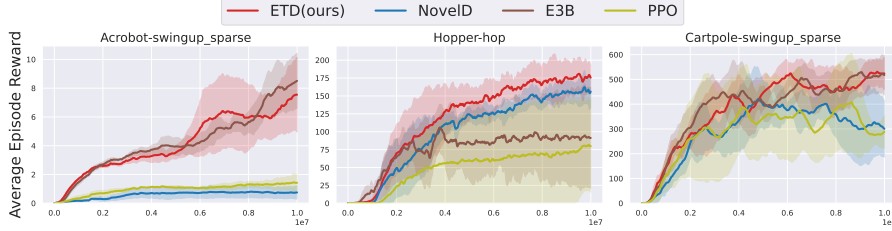

Figure 12: DMC Experiments. All results averaged over 5 seeds.

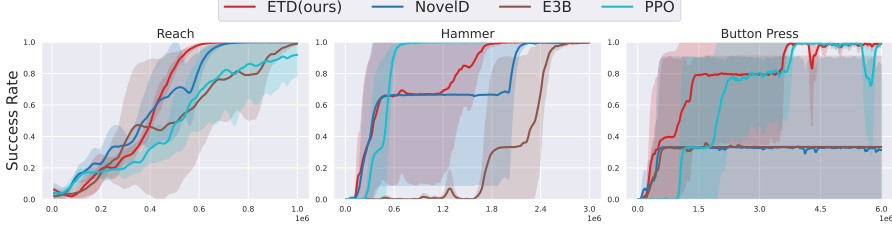

Figure 13: MetaWorld Experiments. All results averaged over 5 seeds.

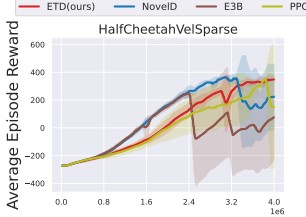

Figure 14: HalfcheetahVelSparse Experiments. All results averaged over 5 seeds.

C.2    ABLATIONS OF ENERGY FUNCTION AND CONTRASITIVE LOSS

Previous work in contrastive RL (Bortkiewicz et al., 2024) has utilized various energy functions and contrastive loss functions for goal-conditioned tasks. We followed their methodology, examining the impact of different energy functions and contrastive losses on temporal distance and performance.

Specifically, we considered four energy functions: Cosine, L2, MRN, and MRN-POT (which we used in this study), defined as follows.

$$
\begin{aligned}
f_{\phi,\cos}(x,y) &= \frac{\langle \phi(x), \phi(y) \rangle}{\|\phi(x)\|_2 \|\phi(y)\|_2}, \\
f_{\phi,l_2}(x,y) &= -\|\phi(x) - \phi(y)\|_2, \\
f_{\phi,\text{MRN}}(x,y) &= -d_\phi(x,y), \\
f_{\phi,\psi,\text{MRN+POT}}(x,y) &= \psi(y) - d_\phi(x,y).
\end{aligned}
\tag{12}
$$

For contrastive loss functions, we evaluated InfoNCE Forward, InfoNCE Backward, InfoNCE Symmetric (which we employed in this work), described below.

$$
\begin{aligned}
\mathcal{L}_{\text{InfoNCE-forward}}(\mathcal{B}; f) &= -\sum_{i=1}^{B} \log \left( \frac{e^{f(x_i, y_i)}}{\sum_{j=1}^{B} e^{f(x_i, y_j)}} \right), \\
\mathcal{L}_{\text{InfoNCE-backward}}(\mathcal{B}; f) &= -\sum_{i=1}^{B} \log \left( \frac{e^{f(x_i, y_i)}}{\sum_{j=1}^{B} e^{f(x_j, y_i)}} \right), \\
\mathcal{L}_{\text{InfoNCE-symmetric}}(\mathcal{B}; f) &= \mathcal{L}_{\text{InfoNCE-forward}}(\mathcal{B}; f) + \mathcal{L}_{\text{InfoNCE-backward}}(\mathcal{B}; f).
\end{aligned}
\tag{13}
$$

Our experiments were conducted on a noisy version of the 17x17 SprialMaze, which is similar to the Disco Maze in (Badia et al., 2020). This version introduced random colors to the walls of the maze, making representation learning more challenging.

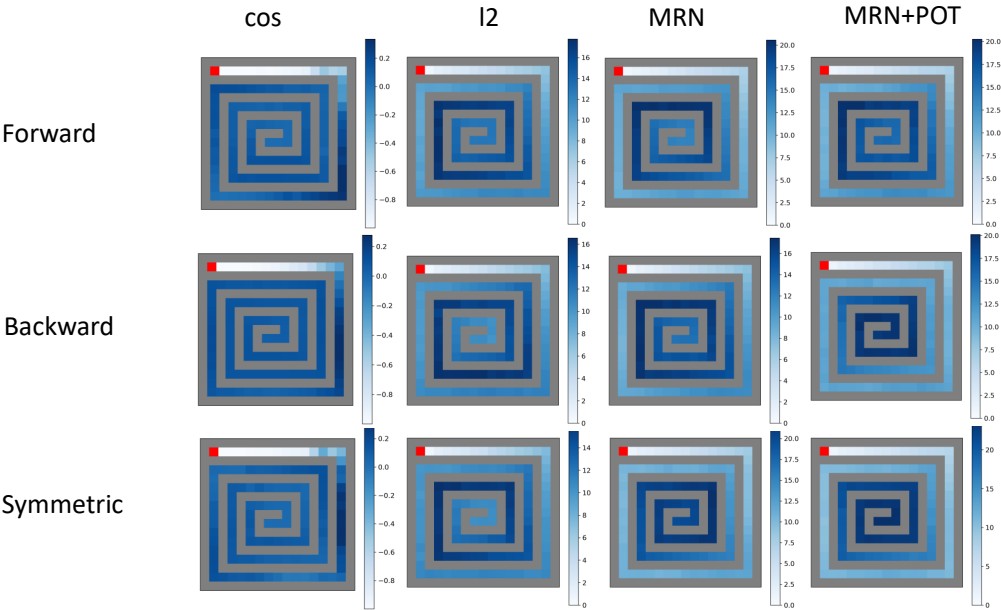

Figure 15: Ablations of Energy Function and Contrastive Loss in 17x17 SpiralMaze-Noisy.

Fig. 15 illustrates the temporal distances learned for each energy function and contrastive loss function. The results indicate that the cosine energy function performed poorly in distinguishing distant states. The L2 and MRN functions produced similar results, while MRN-POT exhibited the

best performance, consistent with our theoretical expectations. For contrastive loss functions, the overall differences were minimal.

Additionally, we conducted exploration tasks in the MiniGrid-ObstructedMaze-1Q environment, as shown in 16. The conclusions from these tasks aligned with the results from the SprialMaze: improved learning of temporal distance correlates with higher exploration efficiency.

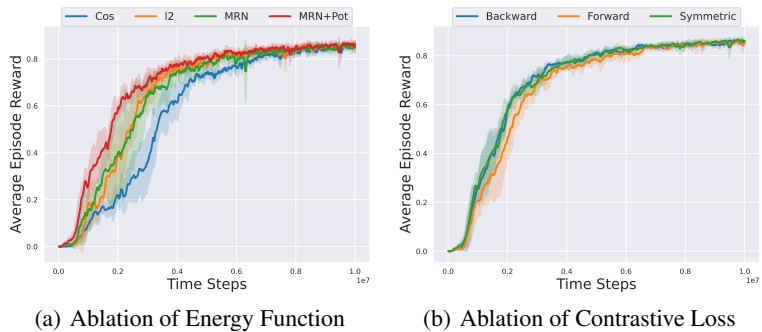

(a) Ablation of Energy Function      (b) Ablation of Contrastive Loss

Figure 16: Ablations of Energy Function and Contrastive Loss in MiniGrid-ObstructedMaze-1Q.

## C.3 MORE COMPLEX EXAMPLES

We add an example of a more complex maze to demonstrate that our temporal distance method can effectively capture the similarity between states. As shown in the Fig. 17, our learned temporal distance remains very close to the ground truth, while inverse dynamics and likelihood methods fail.

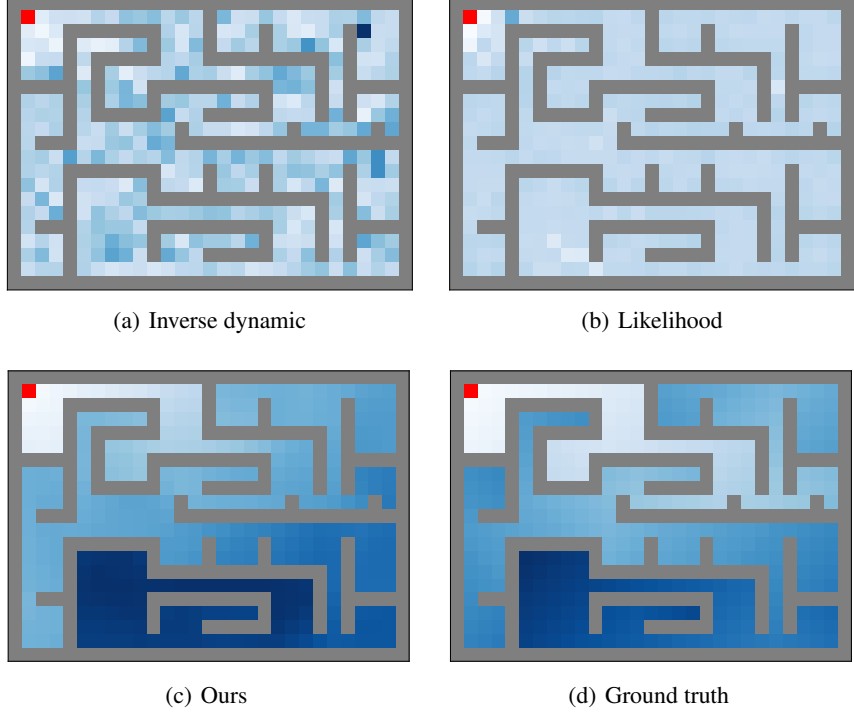

(a) Inverse dynamic      (b) Likelihood

(c) Ours      (d) Ground truth

Figure 17: More Complex Examples

# D    Environment Details

We describe each task environment utilized in our experiments.

## D.1    MiniGrid

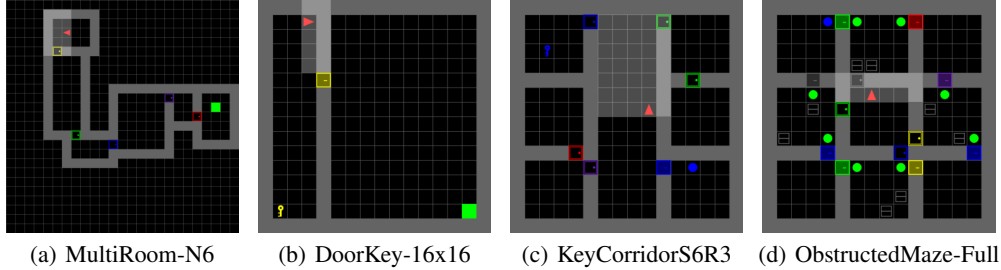

(a) MultiRoom-N6          (b) DoorKey-16x16          (c) KeyCorridorS6R3          (d) ObstructedMaze-Full

Figure 18: Rendering of MiniGrid Environments used in this work.

- **MultiRoom-N6**: An agent initiates from the initial room and navigates towards the goal located in the final room. The rooms are interconnected via a closed yet unlocked door. The variable 'N' denotes the number of rooms.

- **DoorKey-16x16**: The agent is tasked with locating a key in the first room, unlocking and opening a door, and reaching the goal in the second room. 16 refers to the maze size.

- **KeyCorridorS6R3**: The environment comprises a corridor and six rooms separated by walls and doors. The agent, starting from the central corridor, must find a key in an unlocked room and then reach the goal behind another locked door. The prefix 'S' denotes the number of rooms, while 'R' signifies the size of each room. 'S6R3' represents the most complex scenario within the KeyCorridor series of MiniGrid.

- **ObstructedMaze**: The agent's objective is to retrieve a box situated in a corner of a 3x3 maze. Doors are locked, keys are hidden in boxes, and doors are obstructed. The ObstructedMaze is the most challenging environment within MiniGrid, featuring multiple configurations. The notation "NDl" specifies the quantity of locked doors. The presence of a hidden key within a box is denoted by "h," and an obstructed door by a ball is indicated by "b." "Q" signifies the number of quarters that will contain doors and keys out of the nine quarters the map inherently possesses.

  - **ObstructedMaze-2Dlh**: Features 2 locked doors with the key concealed in a box.
  - **ObstructedMaze-2Dlhb**: Includes 2 locked doors, with the key hidden in a box, and a ball obstructing a door.
  - **ObstructedMaze-1Q**: Similar to ObstructedMaze-2Dlhb but with a larger maximum number of steps, and the map consists of a quarter of a 3x3 maze.
  - **ObstructedMaze-2Q**: The map encompasses half of the 3x3 maze.
  - **ObstructedMaze-Full**: The full 3x3 maze.

## D.2    MiniWorld

In this work, we focus on MiniWorld-Maze with different maze sizes, where MazeS3 means the map of maze contains $3 \times 3$ tiles. Figure 19 shows the first-person view observation and the top view of the mazes. In each episode, the agent and goal are initialized at both ends of the diameter of the maze map, which ensures that they are far away enough so that the agent can not easily see the goal without exploration. A random maze will be generated in each episode. There are three possible actions: (1) move forward, (2) turn left, and (3) turn right. The agent will move $0.2 \times tile\_length$ if moving forward, where tile length is the length of one side of a tile. The agent will rotate 90 degrees if turning right/left. The time budget of each episode is $24 \times tile\_num$. The agent will not receive any positive reward if it can not navigate the goal under the time budget.

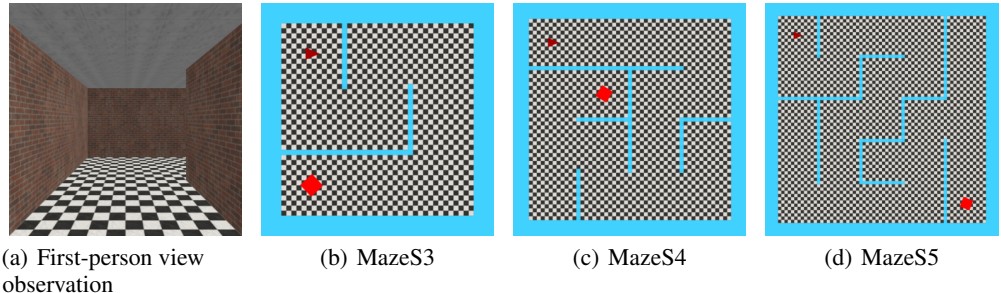

(a) First-person view
observation                (b) MazeS3              (c) MazeS4              (d) MazeS5

Figure 19: Rendering of MiniWorld Environments in this work.

# E    IMPLEMENTATION DETAILS

In the experiments, all methods are implemented based on PPO. We primarily follow the implementation of DEIR*, which is based on Stable Baselines 3 (version 1.1.0).

## E.1    NETWORK STRUCTURES

### E.1.1    POLICY AND VALUE NETWORKS

For the policy and value networks, we follow the definitions of DEIR. All the methods shares the same policy and value network structures.

**MiniGrid**

**CNN**
>           Conv2d(in=3, out=32, kernel=2, stride=1, pad=0),
>           Conv2d(in=32, out=64, kernel=2, stride=1, pad=0),
>           Conv2d(in=64, out=64, kernel=2, stride=1, pad=0),
>           FC(in=1024, out=64).

**RNN**
>           GRU(in=64, out=64).

**MLP (value network)**
>           FC(in=64, out=128),
>           FC(in=128, out=1).

**MLP (policy network)**
>           FC(in=64, out=128),
>           FC(in=128, out=number of actions).

*FC* stands for the fully connected linear layer, and *Conv2d* refers to the 2-dimensional convolutional layer, *GRU* is the gated recurrent units.

**Crafter & MiniWorld**

**CNN**
>           Conv2d(in=3, out=32, kernel=8, stride=4, pad=0),
>           Conv2d(in=32, out=64, kernel=4, stride=2, pad=0),
>           Conv2d(in=64, out=64, kernel=4, stride=1, pad=0),
>           FC(in=576, out=64).

**RNN**
>           GRU(in=64, out=64).

---

*https://github.com/swan-utokyo/deir

**MLP (value network)**
        FC(in=64, out=256),
        FC(in=256, out=1).

**MLP (policy network)**
        FC(in=64, out=256),
        FC(in=256, out=number of actions).

### E.1.2 QUASIMETRIC NETWORK

Our quasimetric network is based on the MRN (Liu et al., 2023). It consists of both a symmetric and an asymmetric component, which together determine the distance between two states. The structure of this network is illustrated in Figure 20. Our potential network shares the same CNN as the quasimetric network, followed by an MLP that outputs a scalar value.

$$d(x,y) = ||h_1(a) - h_2(b)||_2^2 + \max_i\big(h_2(a) - h_2(b)\big)_+[i]$$

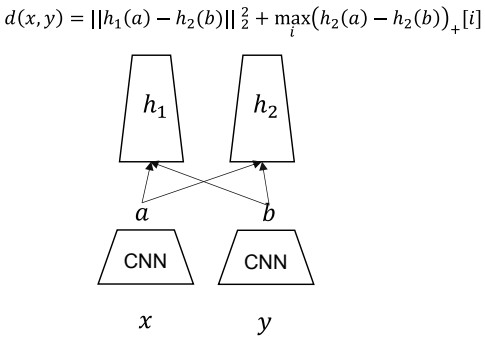

Figure 20: MRN Network

### E.2 HYPERPARAMETERS

We found that applying batch normalization to all non-RNN layers could significantly boost the learning speed, especially in environments with stable observations, a finding also noted in the DEIR paper. We use Adam optimizer with $\epsilon = 1e - 5, \beta_1 = 0.9, \beta_2 = 0.999$. We normalized intrinsic rewards for all methods by subtracting the mean and dividing by the standard deviation.

For ETD, hyperparameters were initially tuned on DoorKey-8x8 and refined using KeyCorridorS6R3 and ObsturctedMaze results. For DEIR, we adopted their original hyperparameters but couldn't fully replicate their ObsturctedMaze-Full performance. Despite this, ETD still outperforms original DEIR performance by a factor of two in sample efficiency in ObsturctedMaze-Full. Our NovelD implementation achieves the best performance reported in the literature. For the Count implementation, we use the episodic form $\mathbb{I}[N_e(s_t) = 1]$ and found it superior to $1/\sqrt{N_e(s_t)}$. The hyperparameters for each method are summarized in following tables. Unless otherwise specified, the hyperparameters are consistent with those used for ETD.

| Hyperparameter | MultiRoom DoorKey KeyCorridor | ObsturctedMaze | Candidate Values |
|---|---|---|---|
| $\gamma$ | 0.99 | 0.99 | / |
| PPO $\lambda_{GAE}$ | 0.95 | 0.95 | / |
| PPO rollout steps | 512 | 512 | / |
| PPO workers | 16 | 16 | / |
| PPO clip range | 0.2 | 0.2 | / |
| PPO training epochs | 4 | 4 | / |
| PPO learning rate | 3e-4 | 3e-4 | / |
| model training epochs | 8 | 4 | 1, 3, 4, 6, 8 |
| mini-batch size | 512 | 512 | / |
| entropy loss coef | 5e-4 | 1e-2 | 5e-4, 1e-2 |
| advantage normalization | yes | yes | / |
| model learning rate | 3e-4 | 3e-4 | 3e-4, 1e-4, 5e-5, 1e-5, 5e-6 |
| normalization for layers | Batch Norm | Layer Norm | Batch Norm, Layer Norm, None |
| extrinsic reward coef | 1.0 | 10.0 | 1, 10 |
| intrinsic reward coef | 1e-2 | 1e-2 | 1e-2, 1e-3, 5e-3, 1e-4 |

Table 2: Hyperparameters for ETD in MiniGrid.

| Hyperparameter | MultiRoom DoorKey KeyCorridor | ObsturctedMaze | Candidate Values |
|---|---|---|---|
| PPO learning rate | 3e-4 | 1e-4 | 3e-4, 1e-4 |
| model training epochs | 4 | 3 | 1, 3, 4, 6, 8 |
| mini-batch size | 512 | 512 | / |
| entropy loss coef | 1e-2 | 1e-2 | 5e-4, 1e-2 |
| model learning rate | 3e-4 | 3e-4 | 3e-4, 1e-4, 5e-5, 1e-5, 5e-6 |
| normalization for layers | Batch Norm | Layer Norm | Batch Norm, Layer Norm, None |
| extrinsic reward coef | 1.0 | 10.0 | 1, 10 |
| intrinsic reward coef | 3e-2 | 3e-3 | 1e-2, 1e-3, 5e-3, 1e-4 |
| $\alpha$ | 0.5 | 0.5 | / |
| $\beta$ | 0 | 0 | / |

Table 3: Hyperparameters for NovelD in MiniGrid.

| Hyperparameter | MultiRoom DoorKey KeyCorridor | ObsturctedMaze | Candidate Values |
|---|---|---|---|
| PPO rollout steps | 512 | 512 | 256, 512 |
| PPO workers | 16 | 64 | 16, 64 |
| PPO learning rate | 3e-4 | 1e-4 | / |
| model training epochs | 4 | 3 | / |
| mini-batch size | 512 | 512 | 512, 2048 |
| entropy loss coef | 1e-2 | 5e-4 | / |
| model learning rate | 3e-4 | 3e-4 | / |
| normalization for layers | Batch Norm | Layer Norm | Batch Norm, Layer Norm, None |
| extrinsic reward coef | 1.0 | 10.0 | / |
| intrinsic reward coef | 1e-2 | 1e-3 | / |
| observation queue size | 1e5 | 1e5 | / |

Table 4: Hyperparameters for DEIR in MiniGrid.

| Hyperparameter | MultiRoom DoorKey KeyCorridor | ObsturctedMaze | Candidate Values |
|---|---|---|---|
| PPO learning rate | 3e-4 | 3e-4 | / |
| mini-batch size | 512 | 512 | / |
| entropy loss coef | 1e-2 | 5e-4 | / |
| normalization for layers | Batch Norm | Layer Norm | Batch Norm, Layer Norm, None |
| extrinsic reward coef | 1.0 | 10.0 | / |
| intrinsic reward coef | 1e-2 | 1e-3 | / |

Table 5: Hyperparameters for Count in MiniGrid.

| Hyperparameter | MultiRoom DoorKey KeyCorridor | ObsturctedMaze | Candidate Values |
|---|---|---|---|
| PPO learning rate | 3e-4 | 3e-4 | / |
| mini-batch size | 512 | 512 | / |
| entropy loss coef | 1e-2 | 1e-2 | / |
| normalization for layers | Batch Norm | Layer Norm | Batch Norm, Layer Norm, None |
| extrinsic reward coef | 1.0 | 10.0 | / |
| intrinsic reward coef | 1e-2 | 1e-2 | / |
| $\alpha$ | 1 | 1 | / |
| $\beta$ | 1 | 1 | / |
| $F$ | 90-th percentil | 1 | / |

Table 6: Hyperparameters for EC in MiniGrid.

| Hyperparameter | MultiRoom DoorKey KeyCorridor | ObsturctedMaze | Candidate Values |
|---|---|---|---|
| PPO learning rate | 3e-4 | 3e-4 | / |
| mini-batch size | 512 | 512 | / |
| entropy loss coef | 1e-2 | 1e-2 | / |
| normalization for layers | Batch Norm | Layer Norm | Batch Norm, Layer Norm, None |
| extrinsic reward coef | 1.0 | 10.0 | / |
| intrinsic reward coef | 1e-2 | 1e-2 | / |
| $\lambda$ | 0.1 | 0.1 | / |

Table 7: Hyperparameters for E3B in MiniGrid.

| Hyperparameter | MultiRoom DoorKey KeyCorridor | ObsturctedMaze | Candidate Values |
|---|---|---|---|
| PPO learning rate | 3e-4 | 3e-4 | / |
| mini-batch size | 512 | 512 | / |
| entropy loss coef | 1e-2 | 1e-2 | / |
| normalization for layers | Batch Norm | Layer Norm | Batch Norm, Layer Norm, None |
| extrinsic reward coef | 1.0 | 10.0 | / |
| intrinsic reward coef | 3e-3 | 1e-2 | / |

Table 8: Hyperparameters for RND in MiniGrid.

| Hyperparameter | ETD | NovelD | DEIR |
|---|---|---|---|
| $\gamma$ | 0.99 | 0.99 | 0.99 |
| PPO $\lambda_{GAE}$ | 0.95 | 0.95 | 0.95 |
| PPO rollout steps | 512 | 512 | 512 |
| PPO workers | 16 | 16 | 16 |
| PPO clip range | 0.2 | 0.2 | 0.2 |
| PPO training epochs | 4 | 4 | 4 |
| PPO learning rate | 3e-4 | 3e-4 | 3e-4 |
| model training epochs | 4 | 4 | 4 |
| mini-batch size | 512 | 512 | 512 |
| entropy loss coef | 1e-2 | 1e-2 | 1e-2 |
| advantage normalization | yes | yes | yes |
| model learning rate | 1e-4 | 1e-4 | 1e-4 |
| normalization for layers | Layer Norm | Layer Norm | Layer Norm |
| extrinsic reward coef | 1.0 | 1.0 | 1.0 |
| intrinsic reward coef | 1e-2 | 1e-2 | 1e-2 |
| $\alpha$ | / | 0.5 | / |
| $\beta$ | / | 0 | / |
| observation queue size | / | / | 1e5 |

Table 9: Hyperparameters for ETD, NovelD and DEIR in Crafter.

| Hyperparameter | ETD | NovelD | DEIR |
|---|---|---|---|
| $\gamma$ | 0.99 | 0.99 | 0.99 |
| PPO $\lambda_{GAE}$ | 0.95 | 0.95 | 0.95 |
| PPO rollout steps | 512 | 512 | 512 |
| PPO workers | 16 | 16 | 16 |
| PPO clip range | 0.2 | 0.2 | 0.2 |
| PPO training epochs | 4 | 4 | 4 |
| PPO learning rate | 3e-4 | 3e-4 | 3e-4 |
| model training epochs | 16 | 4 | 4 |
| mini-batch size | 512 | 512 | 512 |
| entropy loss coef | 1e-2 | 1e-2 | 1e-2 |
| advantage normalization | yes | yes | yes |
| model learning rate | 1e-4 | 1e-4 | 1e-4 |
| normalization for layers | Layer Norm | Layer Norm | Layer Norm |
| extrinsic reward coef | 10.0 | 1.0 | 1.0 |
| intrinsic reward coef | 1e-2 | 1e-2 | 1e-2 |
| $\alpha$ | / | 0.5 | / |
| $\beta$ | / | 0 | / |
| observation queue size | / | / | 1e5 |

Table 10: Hyperparameters for ETD, NovelD and DEIR in MiniWorld.

| Hyperparameter | ETD | NovelD | E3B |
|---|---|---|---|
| $\gamma$ | 0.99 | 0.99 | 0.99 |
| PPO $\lambda_{GAE}$ | 0.95 | 0.95 | 0.95 |
| PPO rollout steps | 512 | 512 | 512 |
| PPO workers | 16 | 16 | 16 |
| PPO clip range | 0.2 | 0.2 | 0.2 |
| PPO training epochs | 4 | 4 | 4 |
| PPO learning rate | 3e-4 | 3e-4 | 3e-4 |
| model training epochs | 4 | 4 | 4 |
| mini-batch size | 512 | 512 | 512 |
| entropy loss coef | 1e-2 | 1e-2 | 1e-2 |
| advantage normalization | yes | yes | yes |
| model learning rate | 3e-4 | 3e-4 | 3e-4 |
| normalization for layers | Layer Norm | Layer Norm | Layer Norm |
| extrinsic reward coef | 1.0 | 1.0 | 1.0 |
| intrinsic reward coef | 1e-2 | 1e-2 | 1e-2 |
| $\alpha$ | / | 0.5 | / |
| $\beta$ | / | 0 | / |
| $\lambda_{E3B}$ | / | / | 0.1 |

Table 11: Hyperparameters for ETD, NovelD and E3B in DMC, MetaWorld and HalfCheetahVec-Sparse.

