# OpenReview forum: "Episodic Novelty Through Temporal Distance"
_ICLR.cc/2025/Conference — ICLR 2025 Poster_

### Official Review · Reviewer_akpb · 2024-11-03

**Soundness:** 3
**Presentation:** 3
**Contribution:** 2
**Rating:** 5
**Confidence:** 3

**Summary:**

This paper addresses the challenge of exploration in sparse reward environments within Contextual Markov Decision Processes (CMDPs), where environments vary between episodes. The authors propose Episodic Novelty Through Temporal Distance (ETD), a method that leverages temporal distance as a metric for state similarity and intrinsic reward computation. Using contrastive learning, ETD estimates temporal distances to assign intrinsic rewards based on the novelty of states within an episode. Experimental results on diverse benchmark tasks show that ETD outperforms state-of-the-art methods, enhancing exploration in sparse reward CMDPs. The paper is well-structured and detailed, and the ETD method is well-described.

**Strengths:**

The paper compares the proposed method with various existing approaches (including DEIR, NovelD, E3B, EC, NGU, RND, and the Count-based method), covering many classical and common algorithms in the exploration field. By testing these methods across multiple benchmark environments, the paper demonstrates the advantages of the ETD method in sparse reward and high-dimensional pixel environments, validating its applicability across different exploration demands and tasks.

**Weaknesses:**

1. The core approach of ETD primarily builds upon the main idea of "Learning Temporal Distances: Contrastive Successor Features Can Provide a Metric Structure for Decision-Making," using contrastive learning to estimate temporal distance. It lacks substantial algorithmic innovation.
The innovation of the ETD (Episodic Temporal Distance) method primarily lies in its approach to learning temporal distances. However, this method closely follows key aspects of the referenced work, "Learning Temporal Distances: Contrastive Successor Features Can Provide a Metric Structure for Decision-Making," in several crucial areas:
1) Discounted State Occupancy Measure: In the ETD method, the discounted state occupancy measure p^\pi_\gamma(s_f = y | s_0 = x) is used to estimate the transition probability from state x to y . The formula is as follows: p^\pi_\gamma(s_f = y | s_0 = x) = (1 - \gamma) \sum_{k=0}^{\infty} \gamma^k p^\pi(s_k = y | s_0 = x) This formula originates from "Learning Temporal Distances," where the discounted state occupancy measure p^{\pi}_{\gamma} is defined as the discounted distribution over future states s^{+} : p^{\pi}_{\gamma}(s^{+} = s' | s_0 = s) = (1 - \gamma) \sum_{k=0}^{\infty} \gamma^k p^{\pi}(s_k = s' | s_0 = s)
2) Successor Distance: The successor distance formula used in ETD is also directly derived from the referenced work, "Learning Temporal Distances."
3) Loss Function: Both works adopt a symmetrized InfoNCE loss function to train the model through contrastive learning, which is a critical component of the approach.
4) Energy Function: The energy function utilized in ETD, f_{\theta = (\phi, \psi)}(x, y) = c_{\psi}(y) - d_{\phi}(x, y) , is also sourced from "Learning Temporal Distances," where it is presented as: f_{\phi, \psi}(s, a, g) = c_{\psi}(g) - d_{\phi}(s, a, g)


2. Experiments are conducted solely on discrete control tasks, with no tests on continuous control tasks (e.g., robot control tasks in Meta-World and Mujoco). State spaces in discrete control tasks are relatively small, allowing agents to explore enough states within a shorter timeframe. In contrast, continuous control tasks present larger state spaces and greater exploration challenges, making it essential to validate the method’s effectiveness in these scenarios. Given that many real-world tasks involve continuous control, this omission limits the method's persuasiveness for broader application scenarios.
3. The absence of publicly available code restricts the ability to independently verify the method's effectiveness.

**Questions:**

1. Please test the performance of ETD on continuous control tasks.
To evaluate the ETD method in continuous control tasks, the authors could consider tasks in environments such as Meta-World (or Panda-Gym) and Mujoco. Examples of basic tasks include "Push," "Reach," and "Pick-Place" in Meta-World (or Panda-Gym), which offer high environmental complexity and sparse reward settings, making them suitable for assessing ETD's exploratory capabilities. Additionally, Mujoco environments like "Cheetah-Vel-Sparse," which have high-dimensional state spaces, could further validate ETD’s performance in high-dimensional continuous spaces. By plotting and comparing learning curves across environments—such as average episode rewards or success rates over training steps—the authors could demonstrate whether ETD converges faster or more stably in sparse reward continuous control tasks compared to other methods.

2. Please provide code.

---

> ### Author Response · Authors · 2024-11-20
> **Authors' Response to Reviewer akpb (1/2)**
>
> We sincerely thank you for your valuable time, constructive criticism, and insightful suggestions to help improve the quality of our paper. In response to your questions and concerns, we have made the following revisions and clarifications.
>
> **Q1: Relationship with “Learning Temporal Distances: Contrastive Successor Features Can Provide a Metric Structure for Decision-Making” and our contribution.**
>
> A: The core intuition behind our method is that providing an intrinsic reward to the agent enhances exploration in sparse reward settings. While existing count-based and similarity-based methods have limitations, using temporal distance addresses both noisy-TV problems and state similarity issues that satisfy the triangle inequality. This is crucial for the Contextual MDP (CMDP) problems with high-dimensional (image) observations that we focus on.
>
> In learning temporal distance, we drew inspiration from "Learning Temporal Distances: Contrastive Successor Features Can Provide a Metric Structure for Decision-Making" (CMD). However, extending this concept to CMDP problems is non-trivial for several reasons:
>
> 1. **Problem Settings:** CMD primarily targets on Goal-conditioned RL (GCRL) problems, whereas our method (ETD) addresses sparse reward CMDP problems.
> 2. **Policy Learning:** CMD's policy minimizes the temporal distance between the current state and the goal directly. In contrast, our method uses this distance as an intrinsic reward to encourage PPO exploration, effectively maximizing the temporal distance between the current state and the episodic history.
> 3. **Definition of Successor Distance:** CMD's definition of successor distance is independent of $\pi$, whereas our successor distance is dependent on $\pi$ and learned in an on-policy manner. We also provide proof that our on-policy form of successor distance still satisfies the quasimetric property, which differs from the original paper.
> 4. **Code Implementation:** We integrated CMD's temporal distance learning with existing CMDP baseline methods and have made our code publicly available on [https://anonymous.4open.science/r/ETD](https://anonymous.4open.science/r/ETD/README.md) to ensure reproducibility.
>
> Furthermore, we conducted comprehensive experiments to validate ETD's performance (Section 5) and performed ablation studies on the aggregate function, asymmetric/symmetric distance function, and the effectiveness of temporal distance itself (Section 5.3).
>
> **Q2: Please provide code.**
>
> A: We have released the code and ensure that all experimental results can be reproduced, see [https://anonymous.4open.science/r/ETD](https://anonymous.4open.science/r/ETD/README.md).

---

> ### Author Response · Authors · 2024-11-20
> **Authors' Response to Reviewer akpb (2/2)**
>
> **Q3: Please test the performance of ETD on continuous control tasks.**
>
> Thank you for your suggestion. We'd like to clarify several key points:
>
> - The scale of state spaces is not directly related to whether action spaces are discrete or continuous. Instead, it's determined by the dimensionality of the state input (e.g. vector or image). In fact, the noisy-TV problem and image-based observations are major reasons why count-based methods don't work well.
>
> - The difficulty of exploration tasks is also not directly related to whether action spaces are discrete or continuous. For instance, in MiniGrid-OMFull, training PPO for 1e7 steps still yields zero cumulative reward. Conversely, in many Meta-World tasks, PPO can achieve optimal performance in just 1e6 steps.
>
> - We acknowledge that continuous action tasks have many real-world applications. However, as baseline methods (RND, NovelD, NGU, E3B, DEIR) for sparse reward CMDP problems primarily focus on discrete control tasks and haven't been implemented for continuous control tasks, we aligned our experiments accordingly.
>
> - To our knowledge, the continuous control tasks mentioned by the reviewer aren't primarily designed for sparse reward exploration problems, and our baseline methods haven't been tested on these tasks. For instance, Meta-World uses dense rewards. Even if we zero out step-wise rewards and only keep terminal success signals, PPO can still learn some tasks within 1e6 steps. For more complex tasks, all current intrinsic motivation methods struggle to complete them. Nevertheless, we did as the Reviewer suggested, selecting tasks from DMC, Meta-World, and MuJoCo “Cheetah-Vel-Sparse” to test ETD and baseline methods in continuous control problems. Results can be found in Appendix B.2.
>
> **DMC Experiments (Figure 12):** We selected three tasks with relatively sparse rewards from the DMC environment: "Acrobot-swingup_sparse", "Hopper-hop", and "Cartpole-swingup_sparse". Our results demonstrate that ETD consistently performs well, showing significant improvements over PPO. We also reproduced two baselines that excelled in discrete tasks—NovelD and E3B—but found they couldn't maintain consistent performance across all three continuous control tasks.
>
> **Meta-World Experiments (Figure 13):** Meta-World environments typically use dense rewards. We modified these to provide rewards only when tasks succeed. We tested on "Reach," "Hammer," and "Button Press" tasks, finding that PPO without intrinsic rewards performed well. This suggests that Meta-World tasks may not be ideal benchmarks for exploration problems. To our knowledge, intrinsic motivation methods haven't been extensively tested on Meta-World, making it less suitable for comparing exploration-based methods.
>
> **MuJoCo Cheetah-Vel-Sparse Experiments (Figure 14):** We couldn't find an official implementation of "Cheetah-Vel-Sparse," and to our knowledge, exploration-based methods haven't been tested in this scenario. However, following the reviewer's suggestion, we modified the Mujoco HalfCheetah environment's forward reward function to provide rewards only when the target velocity is reached. Our experiments revealed that PPO already performs well in this task. Adding intrinsic motivation methods didn't significantly improve performance, likely due to the low exploration difficulty. These environments require relatively low exploration capabilities, do not necessitate intrinsic motivation methods, and are unsuitable as benchmarks for comparing exploration methods.

---

> ### Author Response · Authors · 2024-11-25
> **Looking forward to further comments!**
>
> Dear reviewer akpb,
>
> We have updated our experiments in  DMC, Meta-World, and MuJoCo “Cheetah-Vel-Sparse” in Appendix B.2.  We also made our code publicly available on [https://anonymous.4open.science/r/ETD](https://anonymous.4open.science/r/ETD/README.md). We are wondering if our response and revision have cleared your concerns. We would appreciate it if you could kindly let us know whether you have any other questions. We are looking forward to comments that can further improve our current manuscript. Thanks!
>
> Best regards,
>
> The Authors

---

> > ### Comment · Reviewer_akpb · 2024-11-27
> > **Thanks for the improvement**
> >
> > I have raised my score.
> > BTW, is the crafter score of ETD significantly higher than other methods in Figure 11? It seems the differences are subtle.

---

> > > ### Author Response · Authors · 2024-11-27
> > > **Thanks for the increased score and welcome any concerns and suggestions for improvement.**
> > >
> > > We sincerely appreciate the reviewer's increased score! While our method demonstrated superior performance on MiniGrid and MiniWorld environments, we acknowledge that the results on Crafter were not as significant. We are committed to improving these aspects in our future work and welcome any additional feedback or suggestions.

---

### Official Review · Reviewer_TiSQ · 2024-11-03

**Soundness:** 3
**Presentation:** 3
**Contribution:** 3
**Rating:** 8
**Confidence:** 4

**Summary:**

The paper presents a new intrinsic reward for novelty-based reinforcement learning. The intrinsic reward is based on the estimated temporal distance between two states, which estimates the probability that one state can be reached from another state in multiple time steps. Here, discounting is used to give preference for shorter transitions. The temporal distance is learned by a contrastive loss based on the infoNCE loss. The algorithm is compared against several baseline methods on a wide variety of discrete action exploration tasks and shows promising performance.

**Strengths:**

- A well motivated new intrinsic reward signal for RL
- The concept is simple and provides a strong performance
- Exhaustive evaluations, good comparisons the baseline methods as well as nice ablations
- Good insights why baseline methods fail

**Weaknesses:**

the paper is already of high quality and I could not find major weaknesses. Minor weaknesses are given in the questions section.

**Questions:**

- a more detailed discussion on the successor distance and its assumptions would be insightful. E.g., what happens if the MDP does not allow for loops (which is the case we have to deal with real dynamics, e.g. velocities). Then this distance would be infinite, wouldnt it?
- More insights on the parametrization of the energy function. Why do we need separate networks for  the potential and the assymetric distance? Could we also ablate removing the potential here?
- Why do we need to "symmetrize" the infoNCE loss? Does this imply that transitions are "invertible"? Also here, a ablation and more discussion would be nice.

---

> ### Author Response · Authors · 2024-11-20
> **Authors' Response to Reviewer TiSQ**
>
> Thank you for your positive recommendation of our work and your insightful comments. We greatly appreciate your thoughtful review. Below, we address each of your concerns in detail.
>
> **Q1: Detailed discussion on the successor distance and its assumptions.**
>
> A: Your perspective is insightful. If a state $y$ is unreachable from state $x$ (e.g., $x$ is an absorbing state), then $p^{\pi}(s_f=y|s_0=x)=0$. According to our definition, the successor distance would be infinite in this case, i.e., $d_{\text{SD}}^{\pi}(x,y)=\infty$. This highlights a theoretical limitation of the successor distance concept, as it assumes we're dealing with ergodic MDP problems. We've acknowledged this limitation in the conclusion section of our paper.
>
> **Q2: More insights on the parametrization of the energy function.**
>
> A: We decompose the energy function into a potential function and a distance function to ensure that, theoretically, the optimal solution of the learned distance function aligns with the temporal distance (successor distance) as shown in Proposition 3. Removing the potential function would violate this principle. In practice, we draw inspiration from CRL research[1], which has explored various energy function forms, including cosine similarity[2], dot product[3], negative L1 and L2 distance[4], and quasimetrics[5,6]. For our method ETD, we primarily consider **four energy function forms**: **Cosine, L2, MRN (which omits the potential), and MRN-POT (which we used in this study)**.
>
> We conducted ablation experiments in a 17x17 SpiralMaze (Figure 13). Results show that the cosine energy function struggled to distinguish distant states effectively. The L2 and MRN functions yielded similar outcomes, while MRN-POT demonstrated the best performance, aligning with our theoretical predictions.
>
> Further ablation experiments in Miniworld-ObstructedMaze-1Q corroborate that decomposing the energy function into potential and asymmetric distance forms can achieve superior performance.
>
> [1] Bortkiewicz, Michał, et al. "Accelerating Goal-Conditioned RL Algorithms and Research." *arXiv preprint arXiv:2408.11052* (2024).
>
> [2] Chen, Ting, et al. "A simple framework for contrastive learning of visual representations." *International conference on machine learning*. PMLR, 2020.
>
> [3] Radford, Alec, et al. "Learning transferable visual models from natural language supervision." *International conference on machine learning*. PMLR, 2021.
>
> [4] Hu, Tianyang, et al. "Your contrastive learning is secretly doing stochastic neighbor embedding." *arXiv preprint arXiv:2205.14814* (2022).
>
> [5] Myers, Vivek, et al. "Learning Temporal Distances: Contrastive Successor Features Can Provide a Metric Structure for Decision-Making." *arXiv preprint arXiv:2406.17098* (2024).
>
> [6] Wang, Tongzhou, et al. "Optimal goal-reaching reinforcement learning via quasimetric learning." *International Conference on Machine Learning*. PMLR, 2023.
>
> **Q3: Detailed discussion on the infoNCE loss.**
>
> A: Theoretically, using symmetric InfoNCE loss yields an optimal solution where the denominator depends solely on $y$. This allows our distance function's optimal solution to recover the successor distance. In contrast, using standard InfoNCE loss results in an optimal solution with a denominator of the form $C(x)p_{s_f}(y)$[7]. In practice, however, the performance difference between the two is minimal. Please see our experiments in Appendix B.2 (Figures 13 and 14).
>
> [7] Eysenbach, Benjamin, et al. "Contrastive learning as goal-conditioned reinforcement learning." Advances in Neural Information Processing Systems 35 (2022): 35603-35620.

---

> > ### Comment · Reviewer_TiSQ · 2024-11-25
> > **Response to Rebuttal**
> >
> > Thanks for the insightful answers. As I was already happy with the paper before I will keep my score.

---

> > > ### Author Response · Authors · 2024-11-25
> > > **Gratitude and Best Wishes**
> > >
> > > Thank you for your thoughtful feedback and support. I truly appreciate your time and effort in reviewing my work. Wishing you great success in your research endeavors as well!

---

### Official Review · Reviewer_gFtv · 2024-11-04

**Soundness:** 3
**Presentation:** 3
**Contribution:** 3
**Rating:** 6
**Confidence:** 3

**Summary:**

This paper presents a novel reinforcement learning approach to measure similarities between visited states, using the resulting state distances as a reward bonus to encourage exploration in tasks with sparse rewards. The methodology is technically sound, and the results are promising.

**Strengths:**

- The paper is well-written, with a clear and cohesive narrative. Most technical details are effectively conveyed through illustrative figures and results from intuitive toy tasks.
- The experimental tasks are appropriately chosen, providing sufficient complexity to evaluate the approach.
- The experimental results, along with ablation studies, clearly demonstrate the advantages of the proposed method over other baselines.
- The paper offers a comprehensive analysis and comparison of different technical approaches used by the proposed method and baseline methods, providing valuable insights for readers.

**Weaknesses:**

## MDP assumption
The definition of the intrinsic bonus reward violates the MDP assumption. In section 2, the total reward $r(s_t, a_t, s_{t+1})$ is a decomposed as the environment reward $r_t^e$ plus the weighted bonus $\beta b_t$. However, $b_t$ is a function that depends on the visited states within the episode, which disrupts the definition of total reward and violates the MDP assumption. In this case, the visited states influences the intrinsic reward, potentially harming the policy built upon MDP, as the actions are solely selected by $\pi(a_t|s_t)$ without knowledge of the states' visiting history. Consequently, the same action $a_t$ may yield completely different rewards $r_{t+1}$, depending on whether the $s_{t+1}$ has been visited in the history or not. I believe this is a critical technical issue in this paper, and should be explained thoroughly. At the same time, do other related works using reward bonuses also encounter the same issue?

## Unclear content
The content between lines 186 and 208 is challenging to follow. I suggest the authors restructure this section, providing additional explanations and intuitions. Specifically:
- The meaning of the sentence, "Contrastive learning can estimate the successor distance when positive samples are drawn from the discounted state occupancy measure," is unclear. What does it mean to draw samples from a measure?
- The motivation for using the energy function and its decomposition into the potential net and quasimetric net in practice is unclear. What is this energy function, and why is it applied here?
- The rationale behind using the infoNCE loss also needs further explanation.

## Minor issues
- The SpiralMaze in Figure 2 appears to have a linear structure, which may not effectively demonstrate ETD's advantage in distance learning. A more complex example could better illustrate the method's strengths.
- What is the best practice of choosing the $\beta$? I think it depends on the specific task and is therefore a HP.
- Line 188, there should be a $ds$ in the integral.

**Questions:**

Please address the issues listed in the weaknesses section.

---

> ### Author Response · Authors · 2024-11-20
> **Authors' Response to Reviewer gFtv (1/2)**
>
> Thank you for your strong support in recognizing the novelty of our work. We're also grateful for your insightful comments and questions, which we'll address point-by-point below.
>
> **Q1: Violation of MDP assumption under the definition of episodic intrinsic reward.**
>
> A: Exploration methods primarily aim to reward previously unvisited states, which necessitates remembering already visited ones. Consequently, the definition of intrinsic rewards inherently violates the Markov property. For instance, count-based methods maintain a buffer for historically visited states, resulting in a non-Markovian learned policy. In fact, learning non-Markovian policies is quite common in RL. Policy networks often integrate RNNs to enhance memory capabilities in sequential decision-making problems[1,2]. These RNNs inherently contain information about historical states, further departing from the Markov assumption.
>
> [1] Chen, Lili, et al. "Decision transformer: Reinforcement learning via sequence modeling." Advances in neural information processing systems 34 (2021): 15084-15097.
>
> [2] Bakker, Bram. "Reinforcement learning with long short-term memory." Advances in neural information processing systems 14 (2001).
>
> **Q2: Detailed explanations about the temporal distance learning method.**
>
> A: Thank you for your suggestions, which have helped us improve our manuscript. We've revised the method section, highlighting modifications in blue for clarity. We'll address your three concerns as follows:
>
> - **Relationship between Contrastive Learning and Successor Distance**: We use contrastive learning to estimate the successor distance, following prior work[3]. Consider the joint distribution $p\_\gamma^\pi\left({s}\_{f}=y \mid {s}\_0=x\right)$ over two states $(x,y)$. Let $p\_s(x)$ be the marginal state distribution and $p\_{s_f}(y)=\int\_{s} p\_s(x) p\_\gamma^\pi\left({s}\_{f}=y \mid {s}\_0=x\right)$ be the corresponding marginal distribution over future states. The energy function $f\_{\theta}(x,y)$ assigns higher values to $(x, y)$ tuples from the joint distribution and lower values to those sampled independently from marginal distributions. In practice, we sample state pairs from trajectories as $\left\\{ x_i, y_i | x_i=s_t, y_i=s_{t+\Delta}, \Delta \sim \text{Geom} (1-\gamma) \right\\}_{i=1}^B$, where $(x\_i, y\_i)$ are positive samples and $(x\_i, y\_j)\_{i\neq j}$ are negative samples (Figure 3). With a sufficiently large batch size B, the optimal solution $f\_{\theta^*}$ based on symmetric InfoNCE loss satisfies $f\_{\theta^*}(x,y) = \log \left( \frac{p^\pi\_\gamma(s^f = y | s\_0 = x)}{C \cdot p\_{s_f}(y)} \right)$. Then we can derive $d^{\pi}\_{\text{SD}}(x, y) = f\_{\theta^*}(y,y) - f\_{\theta^*}(x,y)$.
> - **Decomposition of Energy Function:** While the above method derives the successor distance from contrastive learning results, it doesn't guarantee the triangle inequality and other quasimetric properties due to training errors. To ensure $d^{\pi}\_{\text{SD}}(x, y)$ satisfies these properties at the network structure level, we leverage the fact that the optimal solution $f\_{\theta^*}$ can be decomposed into a potential function (dependent only on the future state) minus the successor distance function (Equation 11, Appendix A). Following prior work[3], we parameterize $f\_{\theta^*}$ as a potential net and a quasimetric net: $f\_{\theta=(\phi,\psi)}(x,y) = c\_{\psi}(y)- d_{\phi}(x,y)$. This approach allows us to learn the successor distance directly while optimizing the contrastive learning loss. After training, we discard $c\_{\psi(y)}$ and use $d\_{\phi}(x,y)$ as our temporal distance.
> - **Reason for choosing Symmetric InfoNCE Loss:** Theoretically, using symmetric InfoNCE loss yields an optimal solution where the denominator depends solely on $y$, whereas standard InfoNCE loss results in a denominator of the form $C(x)p\_{s_f}(y)$[4]. In practice, however, the performance difference between the two is minimal, as demonstrated in our experiments in Appendix B.2 (Figures 13 and 14).
>
> [3] Myers, Vivek, et al. "Learning Temporal Distances: Contrastive Successor Features Can Provide a Metric Structure for Decision-Making." *arXiv preprint arXiv:2406.17098* (2024).
>
> [4] Eysenbach, Benjamin, et al. "Contrastive learning as goal-conditioned reinforcement learning." Advances in Neural Information Processing Systems 35 (2022): 35603-35620.
>
> **Q3: More complex example to illustrate ETD’s strengths.**
>
> A: We've included a more complex maze example in Appendix B.3. Figure 16 showcases the results, demonstrating that our method ETD successfully learns temporal distances aligning with the ground truth. In contrast, the other two methods struggle to learn a valid distance in this more challenging environment.

---

> ### Author Response · Authors · 2024-11-20
> **Authors' Response to Reviewer gFtv (2/2)**
>
> **Q4: What is the best practice of choosing the $\beta$?**
>
> A: $\beta$ is a common hyperparameter for methods based on intrinsic rewards, defined as $r_t=r_t^e + \beta b_t$. We summarize the forms of $b_t$ for recent intrinsic motivation methods in Table 1. The selection of $\beta$ is task-specific, typically chosen from several candidate values (1e-2, 1e-3, 5e-3, 1e-4). In our experiments, we found $\beta$=1e-2 to be generally suitable for MiniGrid, Crafter, and MiniWorld tasks. We provide candidate $\beta$ values for our method and all baseline methods in Tables 2–10, labeled as "intrinsic reward coef". In practice, we normalized intrinsic rewards for all methods by subtracting the mean and dividing by the standard deviation. This ensures that the scale of intrinsic rewards doesn't vary significantly across different tasks. Consequently, we recommend using 1e-2 as a default value for $\beta$ when tackling new tasks.

---

> > ### Comment · Reviewer_gFtv · 2024-11-22
> > **Reply to authors' rebuttal**
> >
> > Thank you for your detailed explanation and the updates made to the paper. I appreciate the effort in addressing my concerns regarding Q2-Q4, which have significantly improved the clarity of the work. However, I maintain a different perspective on the MDP issue, which I consider a limitation of the current paper as well as the related prior works [1, 2]. Nevertheless, I continue to hold a **positive** recommendation for this paper, as I believe this limitation can be effectively addressed in future research, potentially leading to a valuable new contribution.
> >
> > I will now outline my arguments and provide a few suggestions for the authors, as follows:
> >
> > - **Definition of MDP in Sutton's book**
> >
> > I double-checked the definition of the Markov Property as introduced in Sutton's RL book [3], Chapter 3.1, page 49. To ensure clarity, I restate it here with adapted time step notations:
> >
> > > In a Markov decision process, the probabilities given by $p$ completely characterize the environment’s dynamics. That is, the probability of each possible value for $s_{t+1}$ and $r_{t}$ depends on the immediately preceding state and action, $s_{t}$ and $a_{t}$, and, given them, not at all on earlier states and actions. This is best viewed as a restriction not on the decision process, but on the state. The state must include information about all aspects of the past agent–environment interaction that make a difference for the future. If it does, then the state is said to have the **Markov property**. We will assume the Markov property throughout this book, though starting in Part II we will consider approximation methods that do not rely on it, and in **Chapter 17 we consider how a Markov state can be efficiently learned and constructed from non-Markov observations**.
> > >
> > This serves as a precise and essential foundation for my arguments.
> >
> > - **Value function learning can be biased**
> >
> > Using this definition, both applying reward bonuses within an episode or globally would indeed violate the MDP assumption. Even if the original RL task is an MDP, introducing a reward bonus that depends on the entire episode history implicitly transforms the task into a POMDP setting. Both the reviewer and the authors appear to agree on this point.
> >
> > Modeling a POMDP task using an MDP-based framework is problematic because key components in the MDP setting, such as the Bellman equation and value bootstrapping, become inaccurate or biased. For instance, the current method utilizes PPO as the backbone algorithm, which relies on an advantage function to scale the likelihood ratio. While the advantage can be computed in different ways, it fundamentally depends on the value function of the current state, $V(s)$. If the state $s$ does not fully encapsulate all relevant information about the environment, the value function will be biased, which can negatively impact performance.
> >
> > This limitation, however, is not unique to the current work and has been present in prior works utilizing reward bonuses, such as NovelD [1] and the work in [2].
> >
> > Despite this issue, I believe that reward bonuses provide significant advantages for exploration in sparse reward settings and help mitigate the impact of the POMDP transformation, as highlighted in all these works.
> >
> > - **Do the LSTM-RL[4] and Decision Transformer[5] violate the MDP setting?**
> >
> > In my opinion, the answer is No.
> >
> > LSTM-RL is specifically designed for POMDP settings and employs a memory-based method to transform POMDP observations into MDP states. These states are represented as the hidden states of the LSTM, which aligns with the techniques described in Chapter 17 of the RL book [3]. This approach ensures that the Markov property is restored by encoding sufficient history into the state representation.
> >
> > The Decision Transformer, on the other hand, uses a sequence model for trajectory learning, where actions are conditioned on a long context that includes past actions, states, and returns-to-go. However, this does not violate the MDP assumption because there are no explicit restrictions in MDPs regarding how actions are chosen. Moreover, the Decision Transformer does not utilize a value function during training. During evaluation, the transition probabilities and rewards are entirely determined by the current state and action, adhering to the definition of MDP.
> >
> >
> > ## References
> >
> > [1] Tianjun Zhang, Huazhe Xu, Xiaolong Wang, Yi Wu, Kurt Keutzer, Joseph E Gonzalez, and Yuandong
> > Tian. Noveld: A simple yet effective exploration criterion. NeurIPS 2021.
> >
> > [2] Mikael Henaff, Minqi Jiang, and Roberta Raileanu. A study of global and episodic bonuses for
> > exploration in contextual mdps. ICML 2023
> >
> > [3] Richard S. Sutton, Andrew G. Barto, Reinforcement Learning: An Introduction.
> >
> > [4] Bakker, Bram. "Reinforcement learning with long short-term memory." NIPS 2001.
> >
> > [5] Chen, Lili, et al. "Decision transformer: Reinforcement learning via sequence modeling." NeurIPS 2021.

---

> > > ### Comment · Reviewer_gFtv · 2024-11-22
> > > **Reply to authors' rebuttal, part 2**
> > >
> > > **Suggestions**
> > > Based on my experience, transforming a POMDP into an MDP setting can significantly enhance the model's performance. This can be effectively achieved in future work through methods such as:
> > >
> > > a) Employing a sequence model architecture for the value function and policy, enabling them to capture the history and make accurate predictions based on it.
> > >
> > > b) Incorporating intermediate embeddings derived from contrastive learning as part of the input to the value function, enriching the state representation and helping the model better account for historical information.
> > >
> > > c) I would also suggest that the authors include a brief limitations section in the paper to discuss the potential risk of POMDP arising from the use of reward bonuses. It is unfortunate that this issue has not been explicitly addressed in previous works that incorporate reward bonuses, and highlighting it here would provide valuable insight for future research.
> > >
> > > Best,
> > >
> > > Reviewer gFtv

---

> > > > ### Author Response · Authors · 2024-11-24
> > > > **Reply to Reviewer gFtv**
> > > >
> > > > A: We sincerely thank Reviewer gFtv for your detailed feedback and constructive suggestions. We are honored to receive your positive recommendation. After carefully considering your comments about the Markov property and reviewing Sutton's RL book, we agree with your perspective and have revised our explanation as follows:
> > > >
> > > > - The formulation of intrinsic reward creates a new reward function $r_t = r_t^e(s_t,a_t) + \beta\cdot b_t(s_{t+1}, s_{0:t})$ that **violates the Markovian property** of MDP, which requires dependence only on $s_t,a_t$. Even if the original RL task is an MDP, introducing a reward bonus that depends on an episodic history implicitly transforms the task into a POMDP setting, leading to bias in value function learning.
> > > > - As Reviewer gFtv correctly points out, LSTM-RL and DT methods **do not violate the Markovian property**.
> > > > - Both intrinsic reward methods and DT learn a strictly **non-Markovian policy**, while LSTM can be viewed as a state-update function (described in Chapter 17 of Sutton's RL book), thus LSTM-RL can be viewed as a Markovian policy on an approximately satisfied Markovian state.
> > > >
> > > > We greatly appreciate Reviewer gFtv's suggestions for future research directions:
> > > >
> > > > - **Employ sequence model architectures**: Currently, our image observations are processed through CNNs and RNNs to extract features before being fed into the PPO policy & value networks. However, exploring alternative sequence model architectures, particularly Transformers, merits further research.
> > > > - **Incorporate embeddings from contrastive learning**: Our contrastive learning (CL) and reinforcement learning (RL) components currently operate independently, with separate feature extraction networks and no gradient sharing. In our initial experiments, we attempted to share a single feature extraction network between CL and RL, which led to unstable results. This instability likely arose from competing gradient updates as the network tried to optimize both CL and RL losses simultaneously. Finding better ways to integrate these components while maintaining training stability could be a promising research direction.
> > > > - **Paper Revision**: We have added a brief limitation paragraph in the Conclusion section to discuss the potential risk of POMDP arising from the use of reward bonuses. We are grateful for the reviewer's thorough feedback that helped strengthen our work.

---

> > > > ### Comment · Reviewer_gFtv · 2024-12-02
> > > >
> > > > Thank you for your reply and for adding the limitation section, which will insightfully guide future research within the RL community.
> > > >
> > > > This work is of overall good quality. My score of **6** reflects only minor technical flaws in the POMDP aspect.

---

### Official Review · Reviewer_uqHy · 2024-11-09

**Soundness:** 3
**Presentation:** 3
**Contribution:** 2
**Rating:** 8
**Confidence:** 4

**Summary:**

This paper introduces Episodic Novelty through Temporal Distance (ETD), a new approach for exploration in Contextual Markov Decision Processes (CMDPs) with sparse rewards. ETD uses temporal distance as a metric for state similarity to compute intrinsic rewards, encouraging agents to explore states that are temporally distant from their episodic history. The authors employ contrastive learning to design a quasimetric that estimates episodic temporal distances. They evaluate  ETD when combined with PPO on various CMDP benchmarks, i.e., MiniGrid, and Crafter (pixel-based observations), and MiniWorld (pixel-based observations).

**Strengths:**

1. The paper introduces a novel method to use temporal distance as a (quasi)metric for state similarity.
2. The paper conducts extensive experiments across multiple CMDP environments, comparing ETD to several baseline methods.
3. ETD + PPO demonstrates robust performance improvements, especially in challenging sparse reward scenarios.
4. Results on extensive experiments across multiple CMDP environments, comparing ETD to several baseline methods have been reported.
5. The paper is well-structured, with clear explanations of the motivation, methods, and results.

**Weaknesses:**

1. The proposed ETD doesn’t take into consideration extrinsic rewards to compute similarity. Intuitively, states with similar rewards could be considered similar in terms of the task objective [1].
2. The approach has been primarily tested on discrete action spaces, and its effectiveness in continuous action domains such as MuJoCo [2], DeepMind Control Suite [3], or Fetch [4] environments remains unexplored.

[1] Agarwal, Rishabh, et. al. “Contrastive behavioral similarity embeddings for generalization in reinforcement learning.” arXiv preprint arXiv:2101.05265 (2021).

[2] https://gymnasium.farama.org/environments/mujoco/

[3] Tassa, Yuval, et al. "Deepmind control suite." arXiv preprint arXiv:1801.00690 (2018).

[4] Plappert, Matthias, et al. "Multi-goal reinforcement learning: Challenging robotics environments and request for research." arXiv preprint arXiv:1802.09464 (2018).

**Questions:**

1. Could the authors shed light on why PPO, an on-policy algorithm, was chosen to be combined with ETD instead of off-policy algorithms such as DQN or SAC? Prior work has used off-policy algorithms extensively for sparse-reward tasks [2] [3].
2. Is ETD straightforward to be extended to continuous action spaces? If so, did the authors run any experiments? If not, what could be possible bottlenecks?
3. Did the authors consider combining temporal distance with reward-based similarity, or explore weighting the temporal distance based on extrinsic rewards? Are there any potential trade-offs or challenges in incorporating extrinsic rewards into the similarity metric that the authors anticipate?

[2] Andrychowicz, Marcin, et al. "Hindsight experience replay." Advances in neural information processing systems 30 (2017).

[3] Co-Reyes, John D., et al. "Evolving reinforcement learning algorithms." arXiv preprint arXiv:2101.03958 (2021).

---

> ### Author Response · Authors · 2024-11-20
> **Authors' Response to Reviewer uqHy**
>
> Thank you for your thoughtful review and insightful questions. We appreciate the time and effort you've put into evaluating our work. Your feedback has been valuable in helping us improve our paper and consider new directions for future research. We'll address each of your questions in detail below.
>
> **Q1: Consider combining temporal distance with (extrinsic) reward-based similarity.**
>
> A: Combining temporal distance with reward-based similarity is an interesting idea. However, our tasks primarily focus on exploration in sparse-reward environments. In these settings, most states yield no rewards, making reward-based similarity calculations unreliable. Nonetheless, this approach could be valuable in other contexts—such as learning generalizable representations, as in the paper you referenced. We appreciate the suggestion and will explore this direction in our future work.
>
> **Q2: Experiments on continuous action spaces.**
>
> A: Thank you for your suggestion. Indeed, ETD can be straightforwardly extended to continuous action spaces. We initially focused on discrete action spaces because exploration-based baseline methods are primarily tested in discrete environments, and we wanted to align with them. Following the reviewer's recommendation, we conducted additional experiments in the following two continuous control tasks.
>
> - **DMC Experiments (Figure 12):** We conducted experiments on three tasks from the DMC environment with relatively sparse rewards: "Acrobot-swingup_sparse", "Hopper-hop", and "Cartpole-swingup_sparse". Our results suggest that ETD consistently performs well, showing significant improvements over PPO. We also implemented two baselines that excelled in discrete tasks—NovelD and E3B—but found they couldn't maintain consistent performance across all three tasks.
>
> - **Meta-World Experiments (Figure 13):** Tasks in Meta-World environments, which primarily involve robotics manipulation (similar to Fetch as the reviewer mentioned), typically use dense rewards. We modified these to provide rewards only upon task completion. Our experiments on "Reach," "Hammer," and "Button Press" tasks showed that PPO without intrinsic rewards performed adequately. This result suggests that Meta-World tasks may not be the most suitable benchmarks for exploration problems.
>
> **Q3: Why combined ETD with PPO instead of off-policy algorithms?**
>
> A: We selected PPO primarily because it serves as the foundation for most prior intrinsic reward methods, enabling straightforward comparisons. Moreover, PPO supports high parallelization of environments, which improves time efficiency. It is also robust to hyperparameters, allowing us to focus on enhancing the exploration mechanism.

---

> ### Author Response · Authors · 2024-11-25
> **Looking forward to further comments!**
>
> Dear reviewer uqHy,
>
> We have updated our experiments in continuous action space in Appendix B.2. We are wondering if our response and revision have cleared your concerns. We would appreciate it if you could kindly let us know whether you have any other questions. We are looking forward to comments that can further improve our current manuscript. Thanks!
>
> Best regards,
>
> The Authors

---

> > ### Comment · Reviewer_uqHy · 2024-11-25
> >
> > Thank you for the explanations and including additional experiments in the paper. The new results are insightful. I also went over the reviews from reviewer "gFtv" and the discussions that followed. The authors' acknowledgment of the limitations of the current intrinsic reward formulation is commendable. Based on the revisions made to the paper, I have updated my score.

---

> > > ### Author Response · Authors · 2024-11-26
> > > **Gratitude and Best Wishes**
> > >
> > > Thank you for your thoughtful feedback and kind support. I sincerely appreciate the time and effort you dedicated to reviewing our work. Wishing you great success and meaningful progress in your research endeavors!

---

### Author Response · Authors · 2024-12-02
**Rebuttal Summary**

We sincerely thank all reviewers for their thorough review and valuable feedback. Here are the key concerns we have addressed:

- **Additional Experiments:**
    - **Continuous action space experiments:** Following suggestions from Reviewers `uqHy` and `akpb`, we tested our method (ETD) across DMC, Meta-World, and MuJoCo environments. Our results show improved exploration in sparse reward settings with continuous actions, though with smaller improvements than in discrete action environments, as these environments were not specifically designed for exploration tasks.
    - **Ablation studies:** Our experiments on ETD's energy function and loss function components confirm that our design choices are more suitable.
    - **More Complex example:** We demonstrated ETD's superior ability to learn ground-truth distances compared to similarity measures used in other intrinsic motivation approaches.
- **Detailed Explanations:**
    - **Method:** We enhanced the clarity of our Method section and addressed questions from Reviewers `gFtv`, `TiSQ`, and `akpb` regarding the energy function, loss function, and other technical details.
    - **Implementation:** We explained our choice of PPO over off-policy algorithms, our rationale for not considering reward-based state similarity measurements,, and our preferred selection for hyperparameter $\beta$.
- **Reproducibility and Limitations:**
    - **Code Release:** Our code is available at [https://anonymous.4open.science/r/ETD](https://anonymous.4open.science/r/ETD/README.md), with full result reproducibility.
    - **Limitations:** We acknowledge that current intrinsic reward-based methods violate the Markov assumption, creating additional challenges. Our successor distance definition also requires the ergodic MDP assumption. We thank Reviewers `gFtv` and `TiSQ` for identifying these limitations, which we will address in future work.

---

### Meta-Review · Area_Chair_iTcw · 2024-12-17

**Metareview:**

**Summary**: The paper introduces Episodic Novelty Through Temporal Distance (ETD), a novel exploration method for Contextual Markov Decision Processes (CMDPs) with sparse rewards. ETD uses temporal distance as a state similarity measure, computed via contrastive learning, to encourage exploration of states that are temporally distant from the episodic history. The method is evaluated on benchmarks like MiniGrid, Crafter, and MiniWorld, showing superior performance over state-of-the-art intrinsic reward methods.

**Strengths**:
- The contributions in the paper are novel. Both reviewers `uqHy` and `gFtv` highlight the paper's key innovation—using temporal distance as a similarity metric—which improves upon previous count-based and similarity-based methods.
- The paper presents comprehensive experiments. `uqHy` mentions the variety of CMDP benchmarks (e.g., MiniGrid, Crafter, MiniWorld), while `gFtv` acknowledges the inclusion of challenging environments. All reviewers commend the ablation studies and comparisons to baselines, which provide deeper insights into the method's performance.
- The method displays significant performance improvements over SotA, particularly in noisy environments, where count-based methods fail.
- The paper is well-written. Reviewers find the writing clear, with a cohesive narrative and intuitive explanations, though some sections required clarification.

**Weaknesses**:
- Limited generalization to continuous action spaces: `uqHy` initially criticized the lack of evaluations on continuous action spaces. In response, the authors provided additional experiments on DMC and Meta-World environments, showing some performance gains compared to baselines but smaller than those on discrete tasks.
- Violation of the Markov assumption: `gFtv` points out that the intrinsic reward depends on the history of visited states, violating the Markov property. While the authors acknowledged this as a common limitation of intrinsic motivation methods, it remains a conceptual issue and limitation of the method.

**Recommendation**: Overall, reviewers agree that the paper is novel, well-executed, and presents valuable contributions to exploration in RL. However, some concerns remain about the generalization to continuous action spaces and the theoretical implications of violating the Markov property. As such, my decision is to accept the paper as a poster.

**Additional Comments On Reviewer Discussion:**

The authors addressed most concerns raised during the discussion period:
- Concerns about generalization to continuous action spaces: The authors added new experiments in DMC and Meta-World environments. Reviewers acknowledged the effort, though results showed smaller improvements. I find the improvements sufficiently high to still merit publication.
- Clarity improvements: The authors revised the Method section for clarity, particularly around the energy function and successor distance learning. I think the clarifications did improve the manuscript's writing quality.
- Concerns about the MDP assumption: The authors clarified that violating the Markov property is inherent in many intrinsic reward methods and cited relevant literature. While I agree with `gFtv` that the concerns about the Markov property remain, I still think the paper merits publication conditioned on having a disclaimer or short discussion on this property violation.

---

### Decision · Program_Chairs · 2025-01-22

Accept (Poster)